# COFormer: Towards Solving General Combinatorial Optimization Problems

## Abstract

Combinatorial Optimization Problems (COP) encompasses a wide range of real-world scenarios. While learning-based methods have achieved notable success on specialized COPs, the development of a unified architecture capable of solving diverse COPs with a single set of parameters remains an open challenge. In this work, we present COFormer, a novel framework that offers significant gains in both efficiency and practicality. Drawing inspiration from the success of next-token prediction in sequence modeling, we formulate the solution process of each COP as a Markov Decision Process (MDP), convert the resulting sequential trajectories into tokenized sequences, and train a transformer-based model on this data. To mitigate the long sequence lengths inherent in trajectory representations, we introduce a CO-prefix design that compactly encodes static problem features. Furthermore, to handle the heterogeneity between state and action tokens within the MDP, we adopt a three-stage learning strategy: first, a dynamic prediction model is pretrained via imitation learning; this model then serves as the foundation for policy generation and is subsequently fine-tuned using reinforcement learning. Extensive experiments across eight distinct COPs and various scales demonstrate COFormer's remarkable versatility, emphasizing its ability to generalize to new, unseen problems with minimal fine-tuning, achieving even few-shot or zero-shot performance. Our approach provides a valuable complement to existing neural methods for COPs that focus on optimizing performance for individual problems[1].

## 1 Introduction

Combinatorial optimization problems (COP) are pivotal in a wide range of real-world applications, including logistics, industrial management, and supply chain optimization (Singh & Rizwanullah, 2022). With the rapid growth of deep learning technologies, solving COP using learning-based methods has garnered increasing attention, giving rise to the field of Neural Combinatorial Optimization (NCO) (Kim et al., 2022; Drakulic et al., 2024b). Among all NCO schemes, the auto-regressive construction methods are favored in the recent literature (Bello et al., 2016; Kool et al., 2018; Kwon et al., 2020; Kim et al., 2022). These methods construct solutions incrementally, and the entire problem-solving process can naturally be framed as a Markov Decision Process (MDP). These end-to-end methods offer significant computational efficiency and flexibility in generating feasible solutions, as they can easily avoid actions that violate constrains within the MDP framework (Kim et al., 2022).

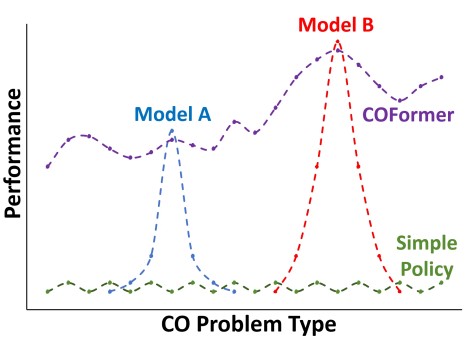

Figure 1: The No Free Lunch Theorem of optimization. While most methods target individual COPs, COFormer solves diverse COPs simultaneously with a single architecture, achieving comparable performance.

---

[1]Our code is available at https://anonymous.4open.science/r/COFormer-35CC/

Figure 2: The difference between previous frameworks and COFormer to solve diverse COPs. While previous frameworks require individual models with specific designs to adapt to different problems, COFormer only utilizes one unified model.

However, models from the existing literature are typically tailored to solve individual COPs, lacking the ability to handle multiple problems simultaneously. This aligns with the No Free Lunch Theorem (NFLT) (Wolpert & Macready, 1997), as most approaches avoid the challenge to achieve generality across different problems, and focus on optimizing performance for specific problems, as illustrated by Model A and Model B in Figure 1. In contrast, using a single unified model across diverse problems offers clear advantages. First, it eliminates the need for hand-crafted network designs for each problem. Second, it enables more efficient adaptation to new COP scenarios compared to training separate models from scratch. Thus, we address the challenge of solving diverse COPs with a unified framework, which leads us to ask: Can a single neural architecture and parameter set simultaneously solve multiple COPs while maintaining strong few-shot capabilities?

Recently, the concept of next-token prediction has marked a new era in general artificial intelligence, excelling in processing data across multiple scenarios, domains, and even modalities. The most successful examples are large language models (LLMs) and multimodal large languange models (MLLMs) (Achiam et al., 2023; Dubey et al., 2024), which can generalize across various natural language process (NLP) and computer vision (CV) scenarios and excel in few-shot learning tasks. Furthermore, the concept has also been applied directly to decision-making tasks (Chen et al., 2021). For instance, a generalist agent was developed to handle different control environments simultaneously, such as Atari games and robot benchmarks (Reed et al., 2022). Motivated by these breakthroughs, we explore whether a single model can be designed to tackle diverse COPs under the same next-token prediction framework.

Building on the auto-regressive MDP formulation, we serialize optimization trajectories from various COPs into flat token sequences, enabling a unified representation space across different problem types. A single Transformer backbone is then trained to learn a universal optimization policy, as shown in Figure 2. Unlike previous NCO methods, which apply auto-regression at the node level, often relying on problem-specific adapters (Drakulic et al., 2024a) or graph symmetries (Kim et al., 2022), our framework operates directly at the universal token level for greater versatility. However, directly applying existing Transformer backbones and training schemes to COPs is inefficient. Due to the NP-hard nature of most COPs, large observation spaces lead to long token sequences and reduced training efficiency. Furthermore, a full trajectory contains different elements, such as states and actions. Predicting all elements uniformly, without accounting for their distinct roles, complicates the training process.

To address these challenges, we propose COFormer, a framework that incorporates two key strategies to enhance training efficiency. First, we introduce a non-causal, decoder-only architecture with a CO-prefix to reduce token sequence length. Unlike dynamic MDP environments, most COPs rely on static features, such as the fixed distances between nodes in the Traveling Salesman Problem (TSP). The CO-prefix aggregates these static features, while the main trajectory handles dynamic elements, significantly reducing sequence length, and improving both training and reasoning efficiency. Second, we decompose the token generation process into separate learning stages. Initially, the model learns to predict forward dynamics from expert trajectories to establish awareness of action consequences, which then serves as a foundation for policy generation. After two rounds of imitation learning (IL), the model is fine-tuned with reinforcement learning (RL) to boost perfor-

mance. This staged approach helps manage the heterogeneous elements in the COP trajectories, reducing overall training complexity.

It is worthy to note that, although several works such as GOAL (Drakulic et al., 2024a) have tackled the challenge of designing general frameworks for COPs, COFormer advances this direction with significantly enhanced versatility. In contrast to GOAL, COFormer does not depend on ad-hoc modules customized for graph-based architectures or learnable adapters designed for specific COPs. Instead, its backbone relies solely on next-token prediction, accommodating arbitrary state representations. This design not only demonstrates greater flexibility and generalization potential but also offers valuable insights for the neural combinatorial optimization (NCO) community.

In summary, our key contributions are as follows:

- We conduct an in-depth exploration of solving multiple COPs via a single unified framework. We believe that this approach provides a valuable complement to existing NCO methods that focus on achieving optimal performance for individual COPs.
- To overcome the challenges of directly applying traditional next-token prediction methods to COPs, we introduce COFormer, a novel framework that incorporates a CO-prefix design and a three-stage learning scheme. This approach effectively reduces token length and mitigates training complexities.
- We establish a comprehensive testbed featuring 8 COPs across various problem scales to evaluate the generic problem-solving ability of our unified CO model. Experiments show that the model exhibits strong generic problem-solving capabilities. Additionally, we demonstrate its few-shot and even zero-shot generalization abilities when tackling new problems, enabled by fast fine-tuning.

## 2 RELATED WORKS

### 2.1 LEARNING-BASED METHODS FOR CO

Solving COPs via learning methods have drawn great attention recently, where many auto-regressive methods are highlighted. The pioneering work in this area was the Pointer Network, which was first tested on TSP(Vinyals et al., 2015). Subsequent research extended this idea by incorporating reinforcement learning (RL), demonstrating its effectiveness across a broader range of COPs (Bello et al., 2016). Routing problems, a significant subclass of COPs, have been extensively studied within this auto-regressive framework using RL (Kool et al., 2018; Kwon et al., 2020). To better account for features at both node and edge levels, a matrix-encoding framework was developed (Kwon et al., 2021). The following literature continues to improve model performances leveraging symmetries of COPs, diversity of learned policies and more advanced search methods (Grinsztajn et al., 2023; Kim et al., 2022; Chalumeau et al., 2023; Hottung et al., 2021; Choo et al., 2022; Ye et al., 2023). The potential of applying auto-regressive NCO methods to more general COPs was also discussed (Drakulic et al., 2024b). These methods offer significant advantages due to their fast inference speed, as their computational complexity during testing remains low. Additionally, they are much more flexible in generating feasible actions that respect various problem constraints.

A recent trend in NCO research is exploring the generalization capabilities of algorithms. Existing methods primarily focus on generalizing across different data distributions (Zhou et al., 2023; Bi et al., 2022) and problem scales (Luo et al., 2023; Zong et al., 2022; Li et al., 2021). In terms of generalization to multiple problems, MVMoE, UniCO and GOAL were proposed as unified frameworks for vehicle routing problems, general TSP and graph-based COPs respectively. In contrast, we seek to develop a unified framework without specific problem type limitations.

### 2.2 NEXT-TOKEN-PREDICTION IN DECISION-MAKING

In addition to the significant success of next-token prediction in both LLMs and MLLMs, researchers have also explored how to directly incorporate it into decision-making problems. (Chen et al., 2021) first explored the use of the Transformer (Vaswani, 2017) as an effective backbone for handling various control environments in an offline RL setting, including Atari, OpenAI Gym, and others. They trained a single policy model to generate actions at each step. (Janner et al., 2021) further proposed the Trajectory Transformer, which predicts all elements within a trajectory. In addition

to offline RL, similar architectures have been integrated with imitation learning (Reed et al., 2022; Shafiullah et al., 2022; Brohan et al., 2022; Zhou et al., 2022). A notable application of this approach is the Generalist Agent, GATO (Reed et al., 2022), which successfully extended its capabilities across multiple control environments using a unified model. (Wen et al., 2022) further adapted the GATO structure, referred to as DB1, and extended it to solve TSP problems. Building on these successes, it is natural to consider Transformers as the backbone for a unified model capable of solving diverse COPs.

However, we note that (Wen et al., 2022) employed a pretrained GCN model (Kipf & Welling, 2016) specifically trained for TSP to generate TSP state embeddings, rather than directly using the original TSP data. We believe that this approach contradicts the core concept of a unified model, which should rely solely on a single architecture and parameter set. Nevertheless, we adopt the unified model structure proposed by GATO and re-implemented by DB1 as a key baseline for comparison, where only the original trajectory data are processed.

## 3 METHODOLOGY

In this section, we first introduce how diverse COPs could be formulated and processed into a unified scheme for further training. Then we introduce the proposed COFormer framework, incorporating the non-causal transformer with CO-prefix design and the three-stage learning approach.

### 3.1 DATA PREPARATION

#### 3.1.1 AUTO-REGRESSIVE MDP FORMULATION FOR CO PROBLEM

We first formulate the sequential construction process of a CO problem solution as an MDP. Following the approach of existing auto-regressive NCO methods (Zhang et al., 2023), a complete solution is incrementally constructed through multiple decision steps.

Let $\mathcal{S}$ denote the entire state space, with states $s_t \in \mathcal{S}$, and let $\mathcal{A} \subseteq \mathcal{S} \times \mathcal{S}$ be the action space, where actions are denoted by $a_t \in \mathcal{A}$. All states are assumed to be reachable from the initial state $s_1$. Since a CO problem is fully observed and deterministic, the transition from state $s_t$ to $s_{t+1}$ is fully determined by action $a_t$. Each state $s_t$ is represented as a set of actions taken before. A policy in the MDP refers to a distribution $P(s'|s)$ over the states $s'$ that can be reached from from $s$ via a single action. A feasible CO problem solution, represented as a complete trajectory $\tau$, can be further induced by the policy over $T$ steps via $\prod_{t=1}^{T} P(s_{t+1}|s_t)$.

It is important to note that *tail recursion* is a common property in COPs: after applying a series of construction steps, the remaining tail subproblem becomes a smaller instance of the original CO problem, as discussed in (Drakulic et al., 2024b). It also includes in particular the Optimality Principle of Dynamic Programming (Bellman, 1954). Any tail-recursive problem can be formulated as the MDP described above, which enables us to further generate the trajectory dataset introduced later.

#### 3.1.2 TRAJECTORY DATASETS

To prepare the trajectory datasets for the IL stages, we first obtain the final optimized solutions from state-of-the-art solvers for various problems. We then trace their complete optimization MDP episodes, $\tau = (\tau_1, \tau_2, ..., \tau_T)$, where each episode consists of states and actions, with $\tau_t = (s_t, a_t)$ representing the state-action pairs at each step.

To jointly handle diverse features from different problems and distributions, we flatten all elements within the MDP episode into one dimension and tokenize them through a tokenization process. Discrete values, such as the node indices of actions, are directly assigned with integer token IDs from $[Min_d, Max_d]$. Continuous values, such as demands and positions, are first encoded via mu-law, discretized into $N_{bin}$ uniform bins, and then tokenized into the range $[Min_c, Max_c]$. The final trajectory token sequence $\overline{\tau}$ at each step is formulated with state tokens, followed by an action spliter token $<|>$, and then action tokens:

$$\overline{\tau} = (\overline{\tau_1}, \overline{\tau_2}, ..., \overline{\tau_T}), \quad \text{where } \overline{\tau_t} = (\overline{s_t}, <|>, \overline{a_t}). \tag{1}$$

Note that the length of a fully tokenized sequence can sometimes be excessively long. To address this, we set the target total token length $L$ in advance, and use selected contiguous segments from complete solution MDPs. Additionally, we only preserve dynamic observations in the intermediate progress within $s_t$, while the static information of the raw problem instances is aggregated within a CO-prefix design, as introduced in the following section. For each problem instance and its complete solution MDP, we collect multiple trajectories as data augmentation. Details of tokenization and trajectory collection can be found in Appendix B.

## 3.2 Non-causal Transformer with CO-prefix

**CO-prefix Mechanism**   Due to the NP-hard nature of most COPs, the observation space can be large, resulting in long token sequences and reduced training efficiency. To tackle this challenge, we decompose the state representation into static and dynamic components, introducing a CO-Prefix which captures the static information and is prepended to the token trajectory as problem metadata. The subsequent sequence then focuses solely on dynamic observations. As illustrated in Figure 6, let $P$ and $\overline{P}$ represent the raw and tokenized CO-Prefix, respectively. The final token sequence fed into the model is $(\overline{P}; \texttt{<X>}, \overline{\tau})$, where $\texttt{<X>}$ denotes a separator token between them. By avoiding repeated encoding of static data (e.g., city coordinates in TSP), this approach dramatically shortens token sequences, which is especially important for Transformer-based models, and offers the following benefits:

- **Reduced training cost:** The CO-Prefix mechanism avoids the $\mathcal{O}(N^2)$ sequence length growth for 8 of 10 COPs, as shown in Table 2, enabling COFormer to scale to very large-scale problems such as TSP and CVRP with $N = 1000$ cities or nodes. More details are provided in Appendix D.1 and Table 5.
- **Faster inference:** For COP solving, computational efficiency requires strict control of context length, making the long-context capability of frontier models less useful.

Although the sequential nature of MDPs makes causal transformers a natural choice, the CO-Prefix is time-invariant and benefits from full bi-directional attention. We therefore adopt a hybrid non-causal transformer where CO-Prefix tokens are processed bi-directionally for richer static representations, while the remaining dynamic tokens are handled causally. This architecture further improves performance on large-scale COPs by combining efficient static encoding with effective sequential modeling. A detailed comparison of the two transformer architectures is provided in Appendix C.

**Action and CO-Prefix Mask**   To ensure that each action selected by COFormer is feasible during inference, we apply an action mask from the MDP environment to filter out actions that violate problem constraints. During training trajectory generation, these masks are recorded alongside the states and later transformed into a CO-Prefix mask, where tokens corresponding to infeasible actions are masked in the attention module. For example, in the TSP the CO-Prefix mask excludes already visited cities, while in the FFSP it excludes job-duration entries of completed tasks. This mechanism focuses the model on feasible tokens without increasing sequence length.

We note that this feasible-action filtering is a common modeling assumption across recent COP methods such as GOAL (Drakulic et al., 2024a) and Sym-NCO (Kim et al., 2022), which also rely on the environment or a search procedure to return valid action candidates at each step. While this assumes locally verifiable constraints and may limit universality when legality depends on global properties, it covers a large class of mainstream COPs and enables efficient training and inference.

While the CO-Prefix design is efficient, COFormer is not strictly dependent on it and can also handle problems without such prefix information, including fully dynamic problems, as demonstrated in our evaluation results. Overall, COFormer is not limited to specific problem types.

## 3.3 Three-Stage Policy Learning

Since a complete trajectory consists of different types of elements, such as observations and actions, predicting them without distinguishing their individual roles further increases the training difficulty. To address this challenge, we decompose the token generation process into three different stages respectively: a dynamics forward IL stage, a policy generation IL stage and a reinforcement finetuning stage, as shown in Figure 3.

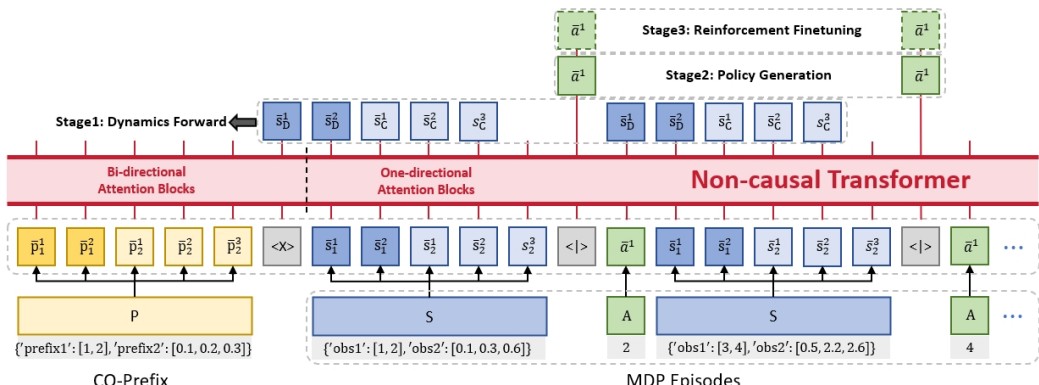

Figure 3: The neural architecture of COFormer. The entire training is separated into three successive stages: the forward dynamics stage, the policy generation stage and the reinforcement finetuning stage.

- **Dynamics forward stage.** In the first stage, we pre-train the model to predict the next observation given the current action, based on the pre-collected expert trajectories. The training loss for a training batch $\mathcal{B}$ is defined as follows:

$$\mathcal{L}(\theta, \mathcal{B}) = -\sum_{b=1}^{|B|} \sum_{t=1}^{T^b} \log p_\theta(\overline{s_{t+1}^b}|(\overline{P^b}, \texttt{<X>}, \overline{\tau_{1:t}^b})), \tag{2}$$

where $T^b$ is the amount of trajectory units in the current token length. Since MDP transitions are deterministic in COPs, thus dynamics model can be accurately trained with the same amount of data. As an auxiliary task to enhance the model's awareness of action consequences, this stage improves the model's capacity to capture local dynamics in token space.

- **Policy generation stage.** In the second stage, we fine-tune the model to generate actions based on the pretrained model in advance, imitating the policy from the expert trajectories. The training loss for a training batch $\mathcal{B}$ is defined as follows:

$$\mathcal{L}(\theta, \mathcal{B}) = -\sum_{b=1}^{|B|} \sum_{t=1}^{T^b} \log p_\theta(\overline{a_{t+1}^b} \mid (\overline{P^b}, \texttt{<X>}, \overline{\tau_{1:t}^b}, \overline{s_{t+1}^b}, \texttt{<|>})) \tag{3}$$

- **Reinforcement finetuning stage.** In the third stage, we further finetune the model to further improve its performance via RL with no expert trajectories involved. We adopt a modified version of GRPO (Shao et al., 2024) as the training algorithm, and the objective function is defined as follows:

$$\mathcal{J}(\theta|\theta_{old}) = \mathbb{E}_{q \sim Q, \{o_i\} \sim \pi_{\theta_{old}}(\cdot|q)}$$

$$\frac{1}{G|o_i|} \sum_{i=1}^{G} \left\{ \min \left[ \frac{\pi_\theta(o_i \mid q)}{\pi_{\theta_{old}}(o_i \mid q)} \hat{A}_i, \ \text{clip}\left( \frac{\pi_\theta(o_i \mid q)}{\pi_{\theta_{old}}(o_i \mid q)}, 1 - \varepsilon, 1 + \varepsilon \right) \hat{A}_i \right] \right\} \tag{4}$$

where $q$ is an instance sampled from a problem class $Q$, $\{o_i\}_{i=1}^{G}$ are the solutions sampled from the old policy $\pi_{\theta_{old}}$, and $\hat{A}_i$ is the advantage, which is computed by employing the greedy solution obtained from the best model so far as baseline.

This three-stage decomposition simplifies the learning process by decomposing the overall process into sub-tasks, allowing the model to first understand intermediate dynamics and then generate qualified policy. This leads to faster and more effective convergence during training.

## 4 PERFORMANCE EVALUATION

### 4.1 PROBLEM AND EXPERT SELECTION

To evaluate the generic problem-solving ability of COFormer, we construct a set of 8 diverse problems for assessment. We first select four common routing problems including Traveling Salesman

Problem (TSP), Vehicle Routing Problem (VRP), Orienteering Problem (OP) and Prize Collecting TSP (PCTSP). To demonstrate how our model handles uncertainty, we also include the stochastic PCTSP (SPCTSP). Beyond routing problems, we evaluate our model on the Knapsack problem. We also include the Flexible Flow Shop Problem (FFSP) (Kwon et al., 2020). Finally, to demonstrate the capability of COFormer on fully dynamic and non-graph based problems, we also include online 3D Bin-packing (3DBP) (Zhao et al., 2021). Details of data generation and token design can be found in the Appendix A.

## 4.2 Evaluation Protocols

**Hyperparameters**    The transformer architecture has 10 layers with 768 embedding dimensions and an input token length of $L = 1000$. During IL training, each epoch consists of 400 batches with 128 samples per batch, uniformly sampled from a mixed set of 8 problems. The RL stage is performed for each single COP with a GRPO group size of 16. We evaluated the model on a separate validation dataset every 2 epochs or GRPO steps and applied early stopping to avoid overfitting. During inference, we use the *KV-Cache* technique modified for the CO-Prefix mechanism to accelerate problem solving. Performance is evaluated on each problem individually using a test dataset of 128 instances per problem, aligned with GOAL (Drakulic et al., 2024a). More implementation details are provided in Table 8 for reproducibility.

**Metrics**    We report four metrics respectively. Following the previous NCO literature, we present the original objectives, the gap from expert results, and the evaluation time on the entire test dataset. Additionally, in line with literature on generic decision-making (Reed et al., 2022), we report performance scores as a percentage, where 100% represents the expert performance for each task, and 0% corresponds to a random policy. The score is calculated as $Score = |obj_e - obj_r|/|obj - obj_r|$, where $obj_e$ and $obj_r$ denote the objectives of the expert and a random policy respectively (Wen et al., 2022).

**Baselines**    We evaluate COFormer with several variations: *COFormer RL* is trained via all three training stages. *COFormer IL* is trained via the two IL stages only without RL finetuning, and *COFormer direct* refers to the model directly trained to generate actions. For baseline comparisons, we first demonstrate the corresponding expert approach for each problem as a straightforward benchmark. We then compare our model with GATO (Reed et al., 2022), which was re-implemented and reported by (Wen et al., 2022) as DB1. Note that we manually implemented the original GATO framework, as it is not open-sourced. Unlike our approach, GATO is trained using a causal transformer structure, where the trajectory data for each problem is prepended with a prompt sequence from the same problem. The prompt consists of multiple step transitions from other episodes, and other key hyperparameters remain the same as ours. Both GATO and COFormer are evaluated using two decoding strategies: greedy decoding and sampling. Finally, we compare COFormer with auto-regressive specialist NCO methods, which also use the MDP formulation for COPs. We report performances on routing problems and Knapsack with AM (Kool et al., 2018), POMO (Kwon et al., 2020), Sym-NCO (Kim et al., 2022) and GOAL (Drakulic et al., 2024a), FFSP with MatNet (Kwon et al., 2021) and 3DBP with PCT (Zhao et al., 2021). Note that MatNet and PCT are used as both the expert solver and the learning baseline for FFSP and 3DBP respectively. All methods that utilize a sampling strategy adopts x16 sampling rollouts.

## 4.3 Performances of Generic Problem Solving

The main evaluation results across all problems are illustrated in Table 1. Due to page limit, we only report problems with N=20 for all routing problems and knapsack, $20 \times 4 \times 3$ for FFSP and 150 packages for 3DBP. The results of N=50 and N=100 instances are shown in Appendix D.1.

**COFormer demonstrates strong generic problem-solving abilities, achieving performance comparable to specialist models.**    With 16-sample decoding, our model achieves scores above 97.00% on all problems except FFSP. Except for CVRP, COFormer outperforms all generic models on all problems. Remarkably, when using sampling, our model even outperforms specialist learning baselines under the same setting on 5 out of the 8 problems.

**The CO-prefix design is significant.** We found that GATO struggles to converge effectively on FFSP, primarily due to its lack of a prefix design. Without this design, GATO computes full obser-

Table 1: Performance results on all COPs with $N = 20$.

| | TSP | | | | Knapsack | | | |
|---|---|---|---|---|---|---|---|---|
| Method | Obj.↓ | Gap↓ | Score↑ | Time↓ | Obj.↑ | Gap↓ | Score↑ | Time↓ |
| Random | 10.47 | - | 0.00% | - | 38.14 | - | 0.00% | - |
| Expert | 3.84 | 0.00% | 100.00% | (2m) | 63.89 | 0.00% | 100.00% | (<1s) |
| POMO no aug | 3.84 | 0.07% | 99.98% | (<1s) | 63.14 | 1.17% | 97.09% | (<1s) |
| Sym-NCO sampling | 3.84 | 0.01% | 99.99% | (1s) | - | - | - | - |
| GOAL multi-task | 3.84 | 0.07% | 99.98% | (5s) | 63.81 | 0.12% | 99.69% | (2s) |
| GATO greedy | 3.99 | 3.80% | 97.68% | (1m) | 62.19 | 2.66% | 93.40% | (27s) |
| GATO sampling | 3.86 | 0.49% | 99.70% | (11m) | 63.53 | 0.56% | 98.60% | (6m) |
| COFormer direct | 3.88 | 1.04% | 99.40% | (<1s) | 61.78 | 3.30% | 91.81% | (<1s) |
| COFormer IL greedy | 3.87 | 0.78% | 99.55% | (<1s) | 61.99 | 2.97% | 92.62% | (<1s) |
| COFormer IL sampling | 3.84 | 0.01% | 99.99% | (7s) | 63.83 | 0.09% | 99.77% | (5s) |
| COFormer RL greedy | 3.84 | 0.07% | 99.98% | (<1s) | 63.53 | 0.56% | 98.60% | (<1s) |
| COFormer RL sampling | 3.84 | 0.01% | 99.99% | (6s) | 63.80 | 0.14% | 99.65% | (6s) |
| | CVRP | | | | OP | | | |
| Method | Obj.↓ | Gap↓ | Score↑ | Time↓ | Obj.↑ | Gap↓ | Score↑ | Time↓ |
| Random | 13.25 | - | 0.00% | - | 1.93 | - | 0.00% | - |
| Expert | 6.11 | 0.00% | 100.00% | (4m) | 5.38 | 0.00% | 100.00% | (1m) |
| AM sampling | 6.29 | 2.96% | 97.40% | (11s) | 5.26 | 2.55% | 95.78% | (5s) |
| Sym-NCO sampling | 6.20 | 1.47% | 98.74% | (1s) | 5.32 | 1.21% | 98.01% | (1s) |
| GOAL multi-task | 6.22 | 1.80% | 98.46% | (5s) | 5.32 | 1.21% | 98.01% | (2s) |
| GATO greedy | 6.63 | 8.51% | 92.72% | (2m) | 4.91 | 8.87% | 85.46% | (40s) |
| GATO sampling | 6.27 | 2.41% | 97.82% | (14m) | 5.30 | 1.56% | 97.42% | (8m) |
| COFormer direct | 6.75 | 10.47% | 91.04% | (<1s) | 5.00 | 7.06% | 88.99% | (<1s) |
| COFormer IL greedy | 6.66 | 9.00% | 92.30% | (<1s) | 5.06 | 5.95% | 90.72% | (<1s) |
| COFormer IL sampling | 6.27 | 2.40% | 97.85% | (11s) | 5.32 | 1.21% | 98.01% | (13s) |
| COFormer RL greedy | 6.34 | 3.76% | 96.78% | (<1s) | 5.16 | 4.09% | 93.62% | (<1s) |
| COFormer RL sampling | 6.26 | 2.38% | 97.96% | (10s) | 5.20 | 3.35% | 94.77% | (13s) |
| | PCTSP | | | | SPCTSP | | | |
| Method | Obj.↓ | Gap↓ | Score↑ | Time↓ | Obj.↓ | Gap↓ | Score↑ | Time↓ |
| Random | 9.25 | - | 0.00% | - | 9.24 | - | 0.00% | - |
| Expert | 3.16 | 0.00% | 100.00% | (2m) | 3.31 | 0.00% | 100.00% | (2m) |
| AM sampling | 3.16 | 0.13% | 99.97% | (9s) | 3.20 | -1.85% | 101.94% | (8s) |
| Sym-NCO sampling | 3.16 | 0.13% | 99.97% | (1s) | - | - | - | - |
| GATO greedy | 3.27 | 3.48% | 98.19% | (1m) | 3.30 | -0.30% | 100.17% | (1m) |
| GATO sampling | 3.20 | 1.26% | 99.36% | (12m) | 3.28 | -0.90% | 100.47% | (12m) |
| COFormer direct | 3.27 | 3.48% | 98.19% | (<1s) | 3.28 | -0.91% | 100.51% | (<1s) |
| COFormer IL greedy | 3.20 | 1.27% | 99.34% | (<1s) | 3.26 | -1.51% | 100.84% | (<1s) |
| COFormer IL sampling | 3.15 | -0.27% | 100.21% | (10s) | 3.16 | -4.03% | 102.89% | (11s) |
| COFormer RL greedy | 3.16 | 0.13% | 99.97% | (<1s) | 3.20 | -1.85% | 101.94% | (<1s) |
| COFormer RL sampling | 3.15 | -0.25% | 100.16% | (11s) | 3.16 | -4.05% | 102.65% | (13s) |
| | 3DBP | | | | FFSP | | | |
| Method | Obj.↑ | Gap↓ | Score↑ | Time↓ | Obj.↓ | Gap↓ | Score↑ | Time↓ |
| Random | 0.1739 | - | 0.00% | - | 45.00 | - | 0.00% | - |
| Expert | 0.8182 | 0.00% | 100.00% | (8s) | 27.31 | 0.00% | 100.00% | (4s) |
| MatNet greedy | - | - | - | - | 27.31 | 0.00% | 100.00% | (4s) |
| PCT | 0.8182 | 0.00% | 100.00% | (8s) | - | - | - | - |
| GATO greedy | 0.7729 | 5.54% | 92.83% | (2m) | 41.42 | 51.67% | 20.24% | (3m) |
| GATO sampling | 0.7831 | 4.29% | 94.55% | (16m) | 41.01 | 50.16% | 22.56% | (1h) |
| COFormer direct | 0.8101 | 0.99% | 98.74% | (16s) | 29.20 | 6.92% | 89.32% | (11s) |
| COFormer IL greedy | 0.8123 | 0.72% | 99.08% | (16s) | 29.11 | 6.59% | 89.82% | (11s) |
| COFormer IL sampling | 0.8152 | 0.37% | 99.53% | (4m) | 28.34 | 3.77% | 94.18% | (2m) |

vation tokens at each step, which becomes highly inefficient when the observation space is large. As a result, GATO can only process few complete trajectory steps in each training episode. The sparse loss signals from action tokens hinder the model's convergence.

**The three-stage learning scheme improves performances.** Compared to COFormer that is directly trained to generate actions, a model fine-tuned on a pre-trained forward dynamics model obtains far better performance across all 8 problems, and the performance gain between *RL-greedy* and *IL-greedy* demonstrates a clear advantage of Reinforcement finetuning stage.

### 4.4 PERFORMANCES ON FEW-SHOT ABILITY

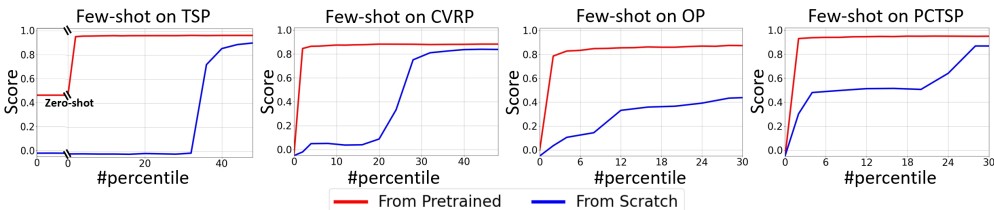

Figure 4: The few-shot results on four routing problems. The x-axis represents the percentage of data used for fine-tuning in relation to the data used in the main results.

To evaluate the few-shot generalization ability of our model on unseen problems, we select 4 routing problems and train four distinct unified models. Each model is trained in a **leave-one-out** manner, excluding the selected problem, and then gradually fine-tuned using datasets from the unseen problem. In each epoch for fine-tuning, we use $0.67\%$ of the total data that was used for the problem in the main results. Figure 4 compares the optimization scores with those of a model trained from scratch, and the fine-tuning costs summarized in Table 6.

Overall, our model demonstrates strong few-shot generalization. In each case, the model achieves high solution quality after just one epoch with minimal data, which shows that COFormer can be quickly adapted to unseen problem, eliminating the need for retraining a separate model and making it well suited for scenarios that require rapid adaptation across related tasks.

Beyond few-shot abilities, we observed even zero-shot generalization on TSP. Using only city coordinates in the prefix and step token designs, which is a subset of other complex routing problems, our pre-trained model can directly generate solutions without additional fine-tuning.

### 4.5 PERFORMANCES WITH RL FINETUNING

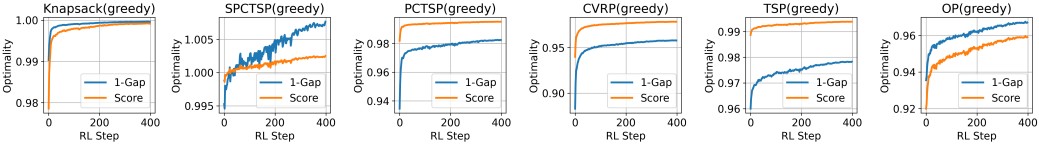

Figure 5: The performance improvement with RL finetuning ($N = 50$).

We also highlight performance gains achieved through RL fine-tuning. Remarkably, COFormer continues to achieve consistent gains in solution quality through RL fine-tuning alone, without relying on any additional expert trajectory data. This indicates that the model is capable of using its own learning signals to refine its optimization policy beyond what was obtained through supervised pre-training. The ability to improve performance without external supervision highlights the potential of COFormer.

### 4.6 FINE-TUNING COSTS AND SCALABILITY

As a foundation model for COPs, COFormer can be fine-tuned to new tasks or larger-scale problems at very low cost, as shown in Table 6 and Appendix D.2. In addition, the model itself is capable of scaling to very large instances such as TSP1000, as reported in Table 5 and Appendix D.1.

## 5 CONCLUSION AND FUTURE WORKS

In this paper, we propose COFormer, a unified model to solve diverse COPs simultaneously. We evaluated the performance of our proposed model on 8 different problems, demonstrating that our approach provides a valuable complement to existing NCO methods that focus on optimizing performance for individual COPs. As for our future work, we plan to enhance our model to tackle problems with significantly longer token sequences and much more diverse data distributions, corresponding to a unified framework with even stronger generalization ability.

REPRODUCIBILITY STATEMENT

To ensure reproducibility, we have provided open source code[2], the dataset construction details are provided in Appendix A, and the training hyperparameters are listed in Table 8.

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

## A   PROBLEM DETAILS

In this section, we continue to introduce the implementation details on each CO problem. We use $N$ to denote either node, item or job amount, and $M$ to denote the total machine amount in FFSP. For each problem, we list the data generation scheme, the expert solver selection, the token (feature) design reference literature, prefix token designs and step token designs respectively. A brief summary is shown in Table 2.

Table 2: The summary of the evaluated COPs, along with individual expert solver to collect trajectories, the prefix token length and the step state token length. $N$ denotes the number of nodes, items, or jobs, depending on the problem, and $M$ denotes the number of machines in the FFSP.

| Problem | Expert Solver | Prefix-Token | State-Token |
|---|---|---|---|
| TSP | LKH3 (Helsgaun, 2017) | $2N$ | 2 |
| MTSP | Google ORTools | $2N + 1$ | 2 |
| VRP | LKH3 (Helsgaun, 2017) | $3N + 2$ | 3 |
| OP | Gurobi (Gurobi Optimization, 2018) | $3N + 2$ | 4 |
| PCTSP | ILS [3] | $4N + 2$ | 3 |
| SPCTSP | ILS | $4N + 2$ | 3 |
| Knapsack | Dynamic Programming | $2N$ | 1 |
| ATSP | LKH3 (Helsgaun, 2017) | $N \times N$ | $N$ |
| FFSP | MatNet (Kwon et al., 2021) | $N \times M$ | $M + 1$ |
| 3DBP | PCT (Zhao et al., 2021) | - | $6 \times (N + 1)$ |

### A.1   TRAVELING SALESMAN PROBLEM (TSP)

In the TSP, the objective is to should find the shortest route that visits each city exactly once and returns to the starting city. The objective is to minimize the total distance of the tour.

**Data Generation:**   We implement the dataset generation scheme described by (Kool et al., 2018), for all TSP instances, the positions of $N$ nodes are uniformly randomly sampled in unit square.

**Expert Solver:**   LKH (Helsgaun, 2017).

**Token (Feature) Design Reference:**   AM (Kool et al., 2018), POMO (Kwon et al., 2020).

**Prefix Tokens:**   Coordinates of each city ($2N$ continuous values).

**Step State Tokens:**   Coordinates of the current city (2 continuous values).

**Step Action Tokens:**   The index of the city to visit next.

### A.2   VEHICLE ROUTING PROBLEM (VRP)

In the Capacitated VRP (Toth & Vigo, 2014), each city has a certain demand. The objective is to construct multiple routes with minimal a distance that all start and end at a given depot, where the total demands of cities within one route should not exceed the capacity limit. Except for the depot, each city should be visited exactly once.

**Data Generation.**   We implement the dataset described by (Nazari et al., 2018). Specifically, each city $i \in \{1, 2, .., N\}$ has a demand $0 < \delta_i \leq D$, where $D > 0$ is the capacity of the vehicle (route). For each route $R_j$, the total demand of the cities along cannot exceed the vehicle's capacity, i.e. $\sum_{i \in R_j} \delta_i \leq D$. For our experiments, We random sample the location coordinates of the depot and the cities within the unit square uniformly. The discrete demands are sampled uniformly from $\{1, 2, ..., 9\}$ and the capacity is set to $D^{20} = 30, D^{50} = 40, D^{100} = 50$.

**Expert Solver**:   LKH (Helsgaun, 2017).

**Token (Feature) Design Reference:**   AM (Kool et al., 2018), POMO (Kwon et al., 2020).

**Prefix Tokens:**   Coordinates of depot and each city ($2(N+1)$ continuous values), demands of each city ($N$ continuous values).

**Step State Tokens:** Coordinates of the current location (2 continuous values), current volume budget(1 continuous value).

**Step Action Tokens:** The index of the location to visit next.

## A.3 ORIENTEERING PROBLEM (OP)

In the OP (Golden et al., 1987), each node is assigned with a specific prize. The objective is to construct a single tour that maximizes the sum of prizes, starting and ending at a give depot. The tour does not have to include every node anymore, but need to be shorter than a length limit.

**Data Generation.** We implement the data generation scheme by (Fischetti et al., 1998; Kool et al., 2018). Specifically, the location coordinates of depot as well as $N$ node are randomly sampled uniformly in the unit square. To make the problem more challenging, we made the prize $p_i$ for each node $i$ proportional to its distance from the depot by setting them as:

$$p_i = 1 + \left\lceil 99 \cdot \frac{d_{0i}}{\max_{j=1}^n d_{0j}} \right\rceil, \hat{p}_i = \frac{p_i}{100}$$

where $d_{0i}$ is the distance from node $i$ to the depot. As for the length limit of the route, we set the fixed maximum length as $T^{20} = 2, T^{50} = 3$ and $T^{100} = 4$, which makes the optimal number of access nodes different from instance to instance.

**Expert Solver:** Gurobi (Gurobi Optimization, 2018).

**Token (Feature) Design Reference:** AM (Kool et al., 2018).

**Prefix Tokens:** Coordinates of the depot and each city ($2(N+1)$ continuous values), prize of each city ($N$ continuous values).

**Step State Tokens:** Coordinates of the current location (2 continuous values), total prize collected so far (1 continuous value), current length budget (1 continuous value).

**Step Action Tokens:** The index of the location to visit next.

## A.4 PRIZE COLLECTING TSP (PCTSP)

In the PCTSP (Balas, 1989), the sum of the total prize is no longer an optimization objective, but a constraint. The objective is to minimize the total route length plus the sum of penalties of unvisited nodes which are given ahead, as well as collecting at least a minimal total prize.

**Data Generation.** We implement the data generation scheme by (Kool et al., 2018). Specifically, as in the OP problem mentioned previously, the location coordinates of the depot and all nodes are randomly sampled uniformly within the unit square. For each node $i$, the associated prize $p_i$ and penalty $\beta_i$ should be carefully balanced. If the penalty is too small, the choice of node is almost entirely determined by the total reward constraint; If the penalty is too large, all nodes are always accessed and the total reward constraint fails. Following the reference (Kool et al., 2018), we set the prize and penalty as:

$$t_i \sim \text{Uniform}(0, 1), \quad \rho_i = t_i \cdot \frac{4}{N}$$

$$\beta_i \sim \text{Uniform}\left(0, 3 \cdot \frac{K^N}{N}\right)$$

where $K^N$ is about half of the trajectory length of the TSP problem with $N$ cities, we roughly set it as $K^{20} = 2, K^{50} = 3, K^{100} = 4$, and the minimum total prize is set to 1 for our experiments.

**Expert Solver:** Iterated Local Search (ILS).

**Token (Feature) Design Reference:** AM (Kool et al., 2018).

**Prefix Tokens:** Coordinates of the depot and each city ($2(N+1)$ continuous values), prize of each city ($N$ continuous values), penalty of each city ($N$ continuous values).

**Step State Tokens:** Coordinates of the current location (2 continuous values), prize-to-go to the minimum required total prize (1 continuous value).

**Step Action Tokens:** The index of the location to visit next.

## A.5 STOCHASTIC PCTSP (SPCTSP)

In the SPCTSP, we show how COFormer performs when dealing with uncertainty. Compared to PCTSP, the expected prize of each node is known before the optimization starts, while the real collected prize can only be revealed after visitation.

**Data Generation.** The data generation for SPCTSP is the sameas in PCTSP, except that we additionally generate the expected prize, which has the same distribution of the real prize.The expert solution algorithm is a modified version of ILS, where the tour is re-optimized iteratively, as suggested by (Kool et al., 2018).

**Expert Solver:** Modified Iterated Local Search (ILS) by suggested (Kool et al., 2018).

**Token (Feature) Design Reference:** AM (Kool et al., 2018).

**Prefix Tokens:** Coordinates of the depot and each city ($2(N + 1)$ continuous values), expected prize of each city ($N$ continuous values), penalty of each city ($N$ continuous values).

**Step State Tokens:** Coordinates of the current location (2 continuous values), prize-to-go to the minimum required total prize (1 continuous value).

**Step Action Tokens:** The index of the location to visit next.

## A.6 KNAPSACK

In the Knapsack problem, a group of items with specific values and volumes are given. The objective is to maximize the total value of items selected without exceeding the total capacity. We designed the problem generation scheme manually and implemented the dynamic programming algorithm for trajectory collection.

**Data Generation.** We implement a manually designed data generation scheme. Specifically, The values $v_i$ of each item $i \in \{1, 2, ..., N\}$ are randomly sampled as:

$$v_i \sim \text{Uniform}(2, 20)$$

To make the problem more challenging, items of higher value should have a larger volume. We further introduce some randomness and set the volume $k_i$ of item $i$ as:

$$k_i = (1 + t)v_i$$

where $t \sim \text{Uniform}(\{-0.5, 0.5\})$, which means we increase or decrease the volume of item $i$ uniformly and randomly. we set the fixed total capacity as $T^{20} = 30, T^{50} = 75$ and $T^{100} = 150$.

**Expert Solver:** Gurobi (Gurobi Optimization, 2018).

**Token (Feature) Design Reference:** POMO (Kwon et al., 2020).

**Prefix Tokens:** Values of all items ($N$ discrete values), volumes of all items ($N$ discrete values).

**Step State Tokens:** Current volume budget (1 discrete values).

**Step Action Tokens:** The index of the newly selected item.

## A.7 FLEXIBLE FLOW SHOP PROBLEM (FFSP)

In the FFSP, $N$ jobs have to be processed in several stages with the same order. Each job in each stage can be handled by a machine from $M$ total machines. The time required for each job at different stages on different machines varies. Each machine can only process at most one job at the same time. The goal is to schedule all jobs so that they can be finished with a minimum of time.

**Data Generation.** We directly adopt the data generation scheme and script provided by (Kwon et al., 2021), where $N = 20, M = 12$. We further implement the corresponding MatNet as the only NCO expert solver in our experiments for trajectory generation.

**Expert Solver:** MatNet (Kwon et al., 2021).

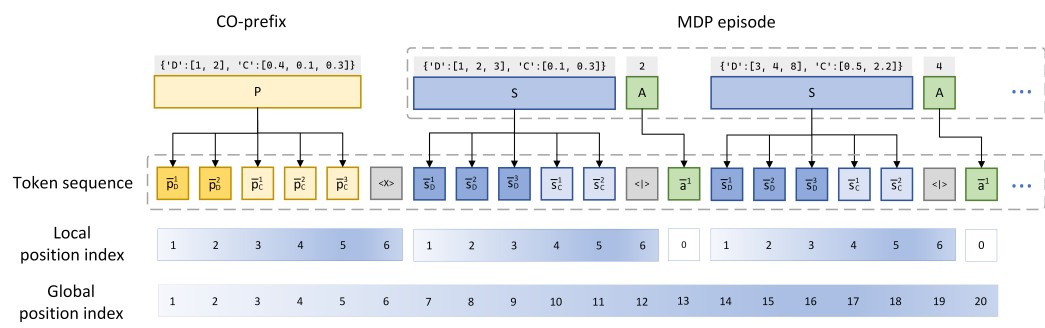

Figure 6: Tokenization illustration of CO-prefix and MDP sequence. 'D' includes all discrete values, and 'C' includes all continuous ones.

**Token (Feature) Design Reference:** MatNet (Kwon et al., 2021).

**Prefix Tokens:** Job durations in each stage on the corresponding machine of each job ($N \times M$ discrete values).

**Step State Tokens:** Job durations of the current machine ($M$ discrete values).

**Step Action Tokens:** The index of the newly selected job to the current machine, or halt.

### A.8 ONLINE 3D BIN-PACKING (3DBP)

In the 3DBP, a set of cuboid-shaped items should be packed. The upcoming items cannot be observed in advance, while only the current item to be packed is observable. The objective is to maximize the total utility rate of all boxes.

**Data Generation.** We directly adopt the data generation scheme and script provided by (Zhao et al., 2021). They use a novel hierarchical representation, packing configuration tree (PCT), to describe the state and action space during packing. We further implement the corresponding PCT approach as the only NCO expert solver in our experiments for trajectory generation.

**Expert Solver:** PCT (Zhao et al., 2021).

**Token (Feature) Design Reference:** PCT (Zhao et al., 2021).

**Prefix Tokens:** None, since 3DBP is fully dynamic and no preliminary information can be presented in advance.

**Step State Tokens:** PCT node configurations.

**Step Action Tokens:** The index of tree node to pack the next item in the current space.

### A.9 MODIFIED TRAVELING SALESMAN PROBLEM (MTSP)

The Modified Traveling Salesman Problem (MTSP) augments the standard TSP with a minimum tour-length constraint and relaxes the visit-once requirement. Given a set of cities and a threshold $L_{\min}$. MTSP asks for a closed tour that starts and ends at the same city, visits every city at least once (allowing revisits), and has minimal length subject to the total tour length being at least $L_{\min}$. MTSP is introduced as a diagnostic COP to investigate whether a model trained on standard routing problems can transfer to instances with altered global constraints and a different feasibility structure.

**Data Generation:** For all MTSP instances, the coordinates of the $N$ nodes are sampled uniformly at random from the unit square, and the minimum-length threshold $L_{\min}$ is drawn uniformly from the set $\{4, 5, 6, 7, 8\}$ for instances with $N = 20$ cities.

**Expert Solver:** Google ORTools.

**Token (Feature) Design Reference:** AM (Kool et al., 2018), POMO (Kwon et al., 2020).

**Prefix Tokens:** Coordinates of each city ($2N$ continuous values), minimum tour-length (1 discrete value)

**Step State Tokens:** Coordinates of the current city (2 continuous values).

**Step Action Tokens:** The index of the city to visit next.

### A.10 ASYMMETRIC TRAVELING SALESMAN PROBLEM (ATSP)

In the asymmetric traveling salesman problem (ATSP), distances between node pairs are no longer determined by Euclidean distances based on node coordinates. Instead, we are given a directed graph with an asymmetric cost matrix, so the distance from node $i$ to node $j$ can differ from the distance from $j$ to $i$. We use ATSP to assess how COFormer performs when handling dense continuous features of complexity $O(N^2)$; in our work, this COP is only used in ablation studies on the $\mu$-law transformation and discretization resolution (see Appendix F).

**Data Generation:** We follow the same data generation scheme as we did for TSP instances. The cities are selected uniformly in a unit square but only adjacency matrix is visible to represent problem instance.

**Expert Solver:** LKH (Helsgaun, 2017).

**Token (Feature) Design Reference:** Raw feature usage.

**Prefix Tokens:** Adjacency matrix ($N \times N$ continuous values), serialized by rows.

**Step State Tokens:** The row of the current city in the adjacency matrix ($N$ continuous values).

**Step Action Tokens:** The index of the city to visit next.

## B TOKENIZATION AND TRAJECTORY COLLECTION DETAILS

In this section, we detail the tokenization and trajectory collection methods used in our model.

### B.1 TOKENIZATION

A complete trajectory sequence fed into our model consists of two components: the CO-prefix and the subsequent transition steps in the corresponding MDP episode, as illustrated in Figure 6. Both raw CO-prefix $P$ and state $s_t$ at each step contain values that can be categorized into discrete and continuous types, as discussed in the previous section. In most COPs, the action representation is a discrete value. Both continuous and discrete values are flattened into a one-dimensional sequence and tokenized separately.

- As for continuous values, our goal is to discretize them and map them to unique token IDs. To achieve this, we use mu-law transformation to convert all values into a fixed range. The mu-law transformation is a common technique to handle continuous signals, ensuring that the values are transformed into a finite range suitable for tokenization. The formula for the mu-law transformation is:

$$F(x) = sgn(x)\frac{log(|x|\mu + 1.0)}{log(M\mu + 1.0)} \tag{5}$$

  where $M = 4$ and $\mu = 15$ in our experiments, and could be adjusted according to different data distribution. The transformed values are further discretized via $N_{bin} = 1800$ bins, and mapped with token IDs of $\mathbb{Z} \in [200, 2000)$.

- As for discrete values, we directly assign them with token IDs from the integer range $\mathbb{Z} \in [0, 200)$. All discrete values encountered in our previous experiments are strictly less than 200, ensuring that this range is sufficient to cover all discrete values in the data.

In addition to the discrete and continuous values, we also introduce two special tokens for separating key parts of the trajectory sequence.

- Action Splitter: The token $<|>$, which separates the state tokens from the action tokens at each step, is assigned the token ID 2000.

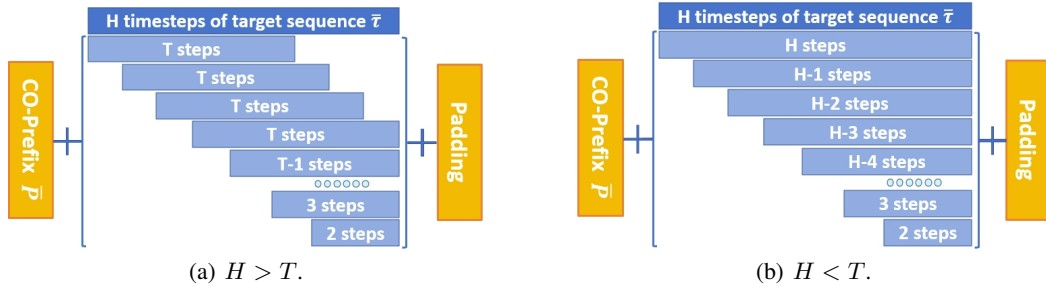

(a) $H > T$.        (b) $H < T$.

Figure 7: Trajectory collection illustration. Instead of directly using all transitions with $T$ steps of the original MDP episode, we collect subsequences and concatenate them to the prefix $\overline{P}$ as the trajectory data we use for training. Trajectory collection illustration.

- Prefix Splitter: The token `<X>`, which separates the CO-prefix from the subsequent MDP episode, is assigned the token ID 2001.

Once the tokens have been assigned, they are embedded into a continuous vector space using a lookup table. This embedding approach, where each token is mapped to a fixed-length vector, is consistent with the methods used in previous works such as (Reed et al., 2022) and (Janner et al., 2021). For position encoding, we employ a combination of both local and global position encodings. The local position encoding uses the local index within each step $\overline{\tau}_t$ or the prefix $\overline{P}$, while the global position encoding follows the traditional approach.

It is important to note that, although several work such as GOAL (Drakulic et al., 2024a) claim to propose general frameworks for COPs, COFormer further advances this direction in the following key ways:

- **Minimal inductive bias**: COFormer does not rely on any graph-formulation assumptions, which guarantees wider applicability.

- **Flexible state representation**: COFormer supports arbitrary key-value dictionaries as state representations without requiring a unified structure or tensor shape, which lowers the engineering cost of adapting new COPs.

- **Sequence-level backbone**: Unlike previous approaches that rely on ad-hoc modules tailored to graph-based models or learnable adapters for specific COP, COFormer aims for a much higher level of generalization. We adopt a novel pipeline inspired by the success of Next-Token Prediction, which does not depend on problem-specific designs. This backbone's success highlights its strong potential, independent of existing NCO methods, and provides valuable insights to the NCO community.

In conclusion, COFormer introduces minimal inductive biases, exhibits the strongest universal properties, and represents the closest approximation to a foundational model for COPs.

### B.2 TRAJECTORY COLLECTION AND DATA AUGMENTATION

In contrast to previous specialist NCO models, which typically use each raw problem instance only once during training or augment it based on symmetries of the CO problem (Kool et al., 2018; Kwon et al., 2020), COFormer employs a different data collection strategy. Each raw problem instance, along with its expert solution trajectory, can be used to generate multiple trajectory data for training, either complete or partial, as illustrated in Figure 7.

We set the target total token length $L$ ($L = 1000$ in our main results) in advance, and compute the length of CO-prefix token length for each instance. The remaining token length, which will be allocated to the trajectory data $\overline{\tau}$, is determined by subtracting the CO-prefix token length from the target total token length $L$. The remaining token length corresponds to the maximum number of time steps $H$ in the target sequence $\overline{\tau}$.

Next, we use the total time steps $T$ from the complete MDP episode and clip subsequences from the original trajectory. If $H > T$, we clip subsequences with steps in the range of $[2, T]$. If $H <= T$, we clip subsequences with steps in the range of $[2, H]$. These subsequences are concatenated to the CO-prefix $\overline{P}$ to form a complete tokenized trajectory. It will be further padded to the target token length $L$, ensuring that each trajectory can be processed in parallel within a batch. The padded tokens are masked during computation so they do not affect model training.

This approach allows for significant data augmentation, as a single problem instance can generate multiple unique trajectories. Importantly, we do not restrict each trajectory to start from its very first time step during training. Instead, the model learns from the internal transitions between various steps in the trajectory, enhancing its ability to generalize across different stages of the solution process.

## C  NON-CAUSAL TRANSFORMER ARCHITECTURE

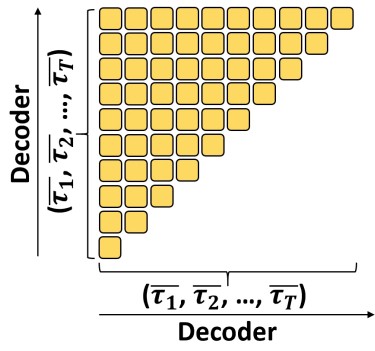
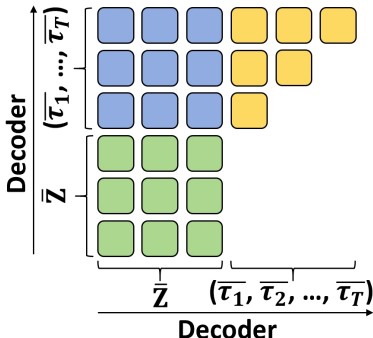

(a) Causal decoder-only architecture without CO prefix.

(b) Non-causal decoder-only architecture with CO-prefix.

Figure 8: Two architecture designs for the unified model. a) Causal decoder-only architecture without CO prefix, where each token is only conditioned on the past tokens and only trajectory data is processed, adopted in (Reed et al., 2022). The entire token length is large. b) Non-causal decoder-only architecture with CO-prefix, where tokens in the CO-prefix shares richer representations conditioned on both prior and past tokens. The trajectory no longer process duplicated static information.

Although the sequential nature of Markov Decision Processes (MDPs) with time-dependent ordering makes the causal transformer architecture a natural choice due to its simple and effective one-directional design, as suggested in previous sequential decision-making literature (Chen et al., 2021; Reed et al., 2022), shown in Figure 8(b), it has certain limitations. Specifically, the CO-prefix $P$ is time-invariant, as it only contains static representations. Therefore, each token within $\overline{P}$ should be fully visible and processed with each other in a bi-directional manner.

To address this, we adopt a non-causal transformer architecture, where the CO-prefix tokens are processed bi-directionally to ensure comprehensive context integration, while the remainder of the sequence is handled in a one-directional manner, as shown in Figure 8(a). The CO-prefix tokens share richer representations, conditioned on both preceding and subsequent tokens, which enhances overall performance.

## D  ADDITIONAL RESULTS

### D.1  SUPPLEMENTARY PERFORMANCES ON LARGER SCALES

In addition to the main results where $N = 20$ for all problems, we further evaluate the performance of COFormer on larger problem scales. Specifically, we examine a problem scale of $N = 50$ and $N = 100$ for routing problems and knapsack, and summarize the results in Table 3 and Table 4.

The results demonstrate that COFormer maintains consistent performance even as the problem scale increases from $N = 20$ to $N = 100$. Notably, our model outperforms the GATO/DB1 baseline and achieves performance comparable to that of single-model baselines. Our model even outperforms the POMO baseline on the Knapsack problem. These results underscore the robustness and scalability of COFormer, confirming that it is capable of handling problem instances with larger scales while maintaining high-quality performance across diverse COPs.

Table 3: Performance results for all routing problems and knapsack with N=50.

| | TSP | | | | Knapsack | | | |
|---|---|---|---|---|---|---|---|---|
| Method | Obj.↓ | Gap↓ | Score↑ | Time↓ | Obj.↑ | Gap↓ | Score↑ | Time↓ |
| Random | 26.08 | - | 0.00% | - | 85.31 | - | 0.00% | - |
| Expert | 5.69 | 0.00% | 100.00% | (2m) | 161.99 | 0.00% | 100.00% | (2s) |
| POMO no aug | 5.73 | 0.70% | 99.80% | (1s) | 161.04 | 0.59% | 98.76% | (1s) |
| Sym-NCO sampling | 5.73 | 0.70% | 99.80% | (1s) | - | - | - | - |
| GOAL multi-task | 5.76 | 1.23% | 99.66% | (7s) | 161.90 | 0.06% | 99.88% | (2s) |
| GATO greedy | 6.25 | 9.86% | 97.22% | (2m) | 160.13 | 0.84% | 97.57% | (2m) |
| GATO sampling | 5.96 | 4.64% | 98.68% | (28m) | 160.63 | 0.81% | 98.20% | (26m) |
| COFormer IL greedy | 5.93 | 4.38% | 98.77% | (1s) | 160.68 | 0.81% | 98.20% | (2s) |
| COFormer IL sampling | 5.78 | 1.45% | 99.59% | (20s) | 161.93 | 0.04% | 99.92% | (22s) |
| COFormer RL greedy | 5.75 | 1.05% | 99.71% | (1s) | 161.80 | 0.12% | 99.73% | (2s) |
| COFormer RL sampling | 5.73 | 0.70% | 99.80% | (22s) | 161.93 | 0.03% | 99.93% | (19s) |
| | CVRP | | | | OP | | | |
| Method | Obj.↓ | Gap↓ | Score↑ | Time↓ | Obj.↑ | Gap↓ | Score↑ | Time↓ |
| Random | 30.67 | - | 0.00% | - | 3.14 | - | 0.00% | - |
| Expert | 10.35 | 0.00% | 100.00% | (9m) | 16.59 | 0.00% | 100.00% | (8m) |
| AM sampling | 10.76 | 3.79% | 98.06% | (7s) | 16.55 | 1.61% | 98.01% | (9s) |
| Sym-NCO sampling | 10.59 | 2.32% | 98.82% | (1s) | 16.26 | 1.99% | 97.55% | (1s) |
| GOAL multi-task | 10.65 | 2.90% | 98.52% | (7s) | 16.26 | 1.99% | 97.55% | (2s) |
| GATO greedy | 11.72 | 12.89% | 93.37% | (2m) | 14.66 | 11.57% | 85.68% | (1m) |
| GATO sampling | 11.19 | 7.87% | 95.96% | (33m) | 15.91 | 4.08% | 94.94% | (18m) |
| COFormer IL greedy | 11.61 | 12.14% | 93.77% | (2s) | 15.49 | 6.64% | 91.77% | (1s) |
| COFormer IL sampling | 11.06 | 6.80% | 96.50% | (31s) | 16.23 | 2.07% | 97.44% | (23s) |
| COFormer RL greedy | 10.93 | 5.45% | 97.21% | (2s) | 15.75 | 4.79% | 94.06% | (1s) |
| COFormer RL sampling | 10.83 | 4.76% | 97.63% | (35s) | 16.11 | 2.88% | 96.42% | (23s) |
| | PCTSP | | | | SPCTSP | | | |
| Method | Obj.↓ | Gap↓ | Score↑ | Time↓ | Obj.↓ | Gap↓ | Score↑ | Time↓ |
| Random | 21.37 | - | 0.00% | - | 21.40 | - | 0.00% | - |
| Expert | 4.48 | 0.00% | 100.00% | (4m) | 4.64 | 0.00% | 100.00% | (4m) |
| AM sampling | 4.53 | 1.15% | 99.69% | (17s) | 4.69 | 1.05% | 99.69% | (17s) |
| Sym-NCO sampling | 4.54 | 1.36% | 99.63% | (1s) | - | - | - | - |
| GATO greedy | 4.92 | 9.89% | 97.27% | (1m) | 5.10 | 9.89% | 97.27% | (1m) |
| GATO sampling | 4.63 | 3.23% | 99.11% | (18m) | 4.79 | 3.23% | 99.11% | (18m) |
| COFormer IL greedy | 4.76 | 6.30% | 98.27% | (2s) | 4.93 | 6.30% | 98.27% | (2s) |
| COFormer IL sampling | 4.54 | 1.36% | 99.63% | (27s) | 4.71 | 1.36% | 99.63% | (27s) |
| COFormer RL greedy | 4.59 | 2.61% | 99.28% | (2s) | 4.64 | -0.14% | 100.07% | (2s) |
| COFormer RL sampling | 4.55 | 1.46% | 99.60% | (26s) | 4.63 | -0.29% | 100.09% | (30s) |

By effectively compressing the sequence length through the CO-Prefix mechanism, COFormer can be further extended to address extremely large-scale problems. We conducted additional experiments on the TSP and CVRP with 1,000 nodes to demonstrate this capability. To ensure training efficiency while keeping the model architecture consistent, we used the same configuration as in Table 1, with the following adjustments:

- Data augmentation was disabled. The model input length was increased to 6,004 for TSP1000 and 8,500 for CVRP1000 to accommodate full MDP trajectories, and the vocab table size for discrete values is increased to 2000 to accommodate a larger range of action tokens.

- The TSP1000 training set consists of 158,000 instances synthesized by DIFUSCO (Sun & Yang, 2023), and the CVRP1000 training set consists of 448,000 instances synthesized by LEHD (Luo et al., 2023). These datasets are used to train *COFormer-direct* to generate actions (stage 2) with a batch size of 240 and an initial learning rate of $5 \times 10^{-4}$ with cosine decay. The *COFormer-RL* model is then trained by fine-tuning from the *COFormer-direct* checkpoint (stages 2 and 3).

Table 4: Performance results for all routing problems and knapsack with N=100.

| | TSP | | | | Knapsack | | | |
|---|---|---|---|---|---|---|---|---|
| Method | Obj.↓ | Gap↓ | Score↑ | Time↓ | Obj.↑ | Gap↓ | Score↑ | Time↓ |
| Random | 52.15 | - | 0.00% | - | 162.53 | - | 0.00% | - |
| Expert | 7.77 | 0.00% | 100.00% | (4m) | 324.73 | 0.00% | 100.00% | (6s) |
| POMO no aug | 7.85 | 1.03% | 99.82% | (<1s) | 324.21 | 0.16% | 99.68% | (<1s) |
| Sym-NCO sampling | 7.84 | 0.90% | 99.84% | (1s) | - | - | - | - |
| GOAL multi-task | 7.85 | 1.03% | 99.82% | (10s) | 324.34 | 0.12% | 99.76% | (3s) |
| GATO greedy | 8.64 | 11.20% | 98.04% | (3m) | 322.02 | 0.83% | 98.33% | (1m) |
| GATO sampling | 8.29 | 6.69% | 98.83% | (42m) | 324.51 | 0.07% | 99.86% | (16m) |
| COFormer IL greedy | 8.16 | 5.02% | 99.12% | (13s) | 323.30 | 0.44% | 99.12% | (2s) |
| COFormer IL sampling | 7.94 | 2.19% | 99.62% | (3m) | 324.56 | 0.05% | 99.90% | (33s) |
| COFormer RL greedy | 7.94 | 2.19% | 99.62% | (13s) | 324.41 | 0.10% | 99.80% | (2s) |
| COFormer RL sampling | 7.92 | 1.93% | 99.66% | (3m) | 324.45 | 0.08% | 99.82% | (31s) |
| | CVRP | | | | OP | | | |
| Method | Obj.↓ | Gap↓ | Score↑ | Time↓ | Obj.↑ | Gap↓ | Score↑ | Time↓ |
| Random | 59.00 | - | 0.00% | - | 3.96 | - | 0.00% | - |
| Expert | 15.65 | 0.00% | 100.00% | (8m) | 33.19 | 0.00% | 100.00% | (3m) |
| AM sampling | 16.44 | 5.05% | 98.18% | (1s) | 31.57 | 4.88% | 94.46% | (1s) |
| Sym-NCO sampling | 15.87 | 1.41% | 99.49% | (1s) | 33.04 | 0.45% | 99.49% | (1s) |
| GOAL multi-task | 16.14 | 3.13% | 98.87% | (10s) | 32.91 | 0.84% | 99.04% | (3s) |
| GATO greedy | 16.77 | 7.16% | 97.42% | (4m) | 30.93 | 6.81% | 92.27% | (2m) |
| GATO sampling | 16.49 | 5.37% | 98.06% | (1h) | 31.79 | 4.22% | 95.21% | (32m) |
| COFormer IL greedy | 16.51 | 5.50% | 98.02% | (22s) | 32.24 | 2.86% | 96.75% | (10s) |
| COFormer IL sampling | 16.01 | 2.30% | 99.17% | (6m) | 32.90 | 0.87% | 99.01% | (3m) |
| COFormer RL greedy | 16.14 | 3.13% | 98.87% | (22s) | 32.71 | 1.44% | 98.66% | (10s) |
| COFormer RL sampling | 16.05 | 2.55% | 99.08% | (6m) | 32.83 | 1.08% | 98.77% | (4m) |
| | PCTSP | | | | SPCTSP | | | |
| Method | Obj.↓ | Gap↓ | Score↑ | Time↓ | Obj.↓ | Gap↓ | Score↑ | Time↓ |
| Random | 41.15 | - | 0.00% | - | 41.12 | - | 0.00% | - |
| Expert | 5.98 | 0.00% | 100.00% | (4m) | 6.22 | 0.00% | 100.00% | (4m) |
| AM sampling | 6.24 | 4.35% | 99.26% | (1s) | 6.32 | 1.61% | 99.71% | |
| Sym-NCO sampling | 5.98 | 0.00% | 100.00% | (1s) | - | - | - | - |
| GATO greedy | 6.61 | 10.54% | 98.21% | (2m) | 7.08 | 13.83% | 97.54% | (2m) |
| GATO sampling | 6.38 | 6.69% | 98.86% | (33m) | 6.64 | 6.75% | 98.80% | (33m) |
| COFormer IL greedy | 6.14 | 2.68% | 99.55% | (10s) | 6.31 | 1.45% | 99.74% | (10s) |
| COFormer IL sampling | 5.98 | 0.00% | 100.00% | (2m) | 6.19 | -0.48% | 100.09% | (2m) |
| COFormer RL greedy | 6.11 | 2.17% | 99.63% | (10s) | 6.22 | 0.00% | 100.00% | (10s) |
| COFormer RL sampling | 5.96 | 0.33% | 100.05% | (2m) | 6.21 | -0.16% | 100.03% | (3m) |

- Evaluation is performed on a single NVIDIA A100 GPU. For TSP1000, the expert (DIFUSCO) uses 10 parallel processes and COFormer uses a batch size of 10 for parallel evaluation. For CVRP1000, both COFormer and the expert (LEHD) are evaluated with a batch size of 5.

As shown in Table 5, COFormer achieves solutions within approximately 4% of a strong expert solver at $N = 1000$, while GATO fails to converge. By separating static features, the CO-Prefix mechanism avoids the $\mathcal{O}(N^2)$ sequence growth caused by repeatedly encoding static information (e.g., TSP city coordinates in each state), achieving a 99.7% reduction in sequence length and thereby enabling the model to handle large instances while maintaining efficient adaptation.

## D.2 GENERALIZATION TO LARGER SCALES

In addition to evaluating COFormer on test sets of the same scale as the training set, we further analyze how well the model generalizes to larger-scale problems. To do so, we use the pre-trained model reported in Table 1 from Section 4 and fine-tune it on newly collected trajectory data for TSP instances with $N = 100$ and $N = 200$. Each fine-tuning run is performed for 10 epochs on 5,250 new problem instances, which takes about 50 minutes, and the results are compared with the POMO baseline (Kwon et al., 2020).

The performance results are shown in Figure 9, illustrating how the model adapts to larger problem sizes. The results show that POMO, as a specialist model, generalizes directly to large-scale prob-

Table 5: Large-scale experiments on the TSP and CVRP problem with N=1000.

| Method | traj. length↓ | Obj.↓ | Gap↓ | Score↑ | Time↓ |
|---|---|---|---|---|---|
| **TSP N=1000** | | | | | |
| Random | - | 520.94 | - | 0.00% | - |
| Expert (DIFUSCO) | - | 23.61 | 0.00% | 100.00% | (3m) |
| GATO | $> 2 \times 10^6$ | - | - | - | - |
| COFormer direct greedy | 6004 | 25.89 | 9.63% | 99.54% | (6m) |
| COFormer direct sampling | 6004 | 24.76 | 4.86% | 99.77% | (102m) |
| COFormer RL greedy | 6004 | 25.41 | 7.62% | 99.64% | (6m) |
| COFormer RL sampling | 6004 | 24.52 | 3.97% | 99.81% | (105m) |
| **CVRP N=1000** | | | | | |
| Random | - | 534.18 | - | 0.00% | - |
| Expert (LEHD) | - | 39.37 | 0.00% | 100.00% | (4m) |
| GATO | $> 3 \times 10^6$ | - | - | - | - |
| COFormer direct greedy | 8500 | 41.66 | 5.88% | 99.53% | (13m) |
| COFormer direct sampling | 8500 | 40.34 | 2.50% | 99.80% | (210m) |
| COFormer RL greedy | 8500 | 40.72 | 3.48% | 99.73% | (13m) |
| COFormer RL sampling | 8500 | 39.90 | 1.41% | 99.89% | (205m) |

Table 6: The Cost of Generalist Fine-Tuning experiments in Figure 4.

| Target COP | Instances/Epoch | Fine-tune time/epoch | Time to 85%+ Score |
|---|---|---|---|
| TSP | 3330 | 10.56m | 32.10m (94.06%) |
| CVRP | 2745 | 6.63m | 13.92m (86.45%) |
| PCTSP | 5095 | 11.23m | 11.90m (93.04%) |
| OP | 6875 | 24.6m | 1.69h (85.11%) |

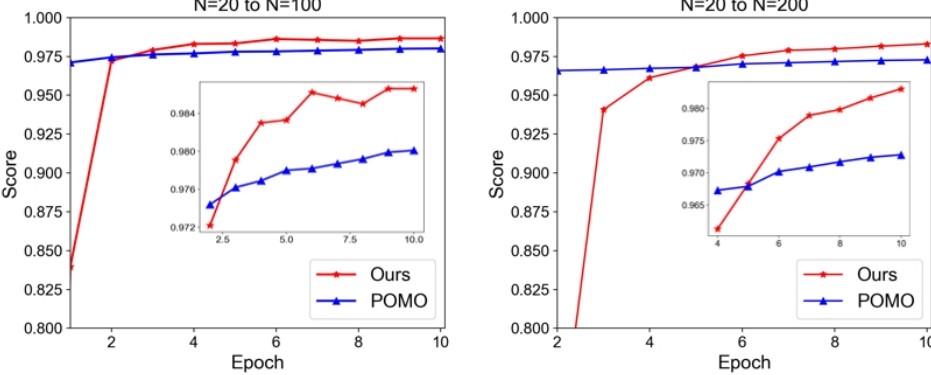

Figure 9: Results of finetuning COFormer trained with $N = 20$ problems in Table 1 to large scale TSP problem with $N = 100$ and $N = 200$.

lems even without fine-tuning, whereas COFormer requires some fine-tuning to recover its problem-solving ability. However, this fine-tuning is fast: a score exceeding 95% is achieved in only 2 epochs (about 10 minutes, 1,050 problem instances), and after 3 and 5 epochs, respectively, CO-Former already outperforms POMO. These results highlight the model's ability to scale effectively and provide valuable insights into the impact of fine-tuning on performance as the problem size increases.

### D.3 ABLATION ON PARAMETER SCALES

To better understand the effect of the parameter scale on overall performance, we train several versions of our model with different parameter scales on five problems with $N = 50$. Specifically, we focus on adjusting the width of the transformer backbone, i.e., the embedding dimensions. The results of these experiments are summarized in Table 7.

Table 7: Ablation study on different embedding dimensions. The best results are in bold.

| | h=128 #params=2.7M | | h=256 #params=9M | | h=512 #params=34M | | h=768 #params=75M | | h=1024 #params=131M | |
|---|---|---|---|---|---|---|---|---|---|---|
| | Obj. | Score | Obj. | Score | Obj. | Score | Obj. | Score | Obj. | Score |
| TSP | 6.82 | 94.44% | 6.02 | 98.37% | 5.94 | 98.78% | 5.96 | 98.66% | **5.92** | **98.83%** |
| CVRP | 12.55 | 89.13% | 11.75 | 93.12% | 11.59 | 93.87% | **11.59** | **93.89%** | 11.68 | 93.41% |
| OP | 11.22 | 60.07% | 15.18 | 89.41% | 15.55 | 92.16% | 15.37 | 90.76% | **15.61** | **92.62%** |
| PCTSP | 5.56 | 93.30% | 4.91 | 97.33% | 4.77 | 98.20% | 4.81 | 98.00% | **4.71** | **98.59%** |
| Knapsack | 140.14 | 69.94% | 160.28 | 97.69% | 160.28 | 97.65% | **160.49** | **97.96%** | 160.36 | 97.80% |

Table 8: Implementation details.

| Module | Element | Detail |
|---|---|---|
| System | OS | Ubuntu 22.04.2 |
| | CUDA | 11.7 |
| | Python | 3.11.4 |
| | Pytorch | 2.0.1 |
| | GPU | $2 \times$ NVIDIA A100 80G |
| | CPU | $2 \times$ Xeon Platinum 8358P |
| Hyperparams | Backbone | Llama (Train from scratch) |
| | Backbone Version | Transformers 4.40.0 |
| | Embedding dimension | 768 |
| | Layer Num | 10 |
| | Q/KV Head Num | 8 |
| | Total token length $L$ | 1000 |
| | RMS Norm epsilon | 1e-6 |
| | Weight Decay | 1e-4 |
| | Early Stopping Runs | 6 |
| | M of $\mu$-law | 4 |
| | $\mu$ of $\mu$-law | 15 |
| | $[Min_d, Max_d)$ | $[0, 200)$ |
| | $[Min_c, Max_c)$ | $[200, 2000)$ |
| | Optimizer | AdamW |
| | inital learning rate | 0 |
| | max learning rate | 2.5e-4 |
| | leanring rate warmup ratio | 5% |
| | leanring rate decay ratio | 75% |
| | leanring rate decay factor | 10 |
| | leanring rate decay style | cosine |
| | RL group size | 16 |
| | RL policy update times per step | 1 |
| | RL clipping threshold ($\epsilon$) | 0.2 |
| | RL leanring rate | 5e-5 |
| | RL leanring rate decay style | cosine |

We observe that the performance of our model continues to improve as the total parameter scale increases. However, the rate of improvement gradually slows down when the total parameter scale reaches 75M and 131M, corresponding to embedding dimensions of 768 and 1024, respectively. Among these configurations, the model with 131M parameters outperforms the model with 75M parameters on 3 out of 5 problems.

While increasing the parameter scale generally improves performance, we find that further scaling the parameters beyond a certain point yields diminishing returns. This suggests that the current limitations are not solely related to parameter scale but may also be influenced by the number of problem types and the amount of data used for training. Moving forward, we aim to further explore how increasing the diversity of problem types and expanding the data size can enhance the scalability of our model, unlocking its full potential.

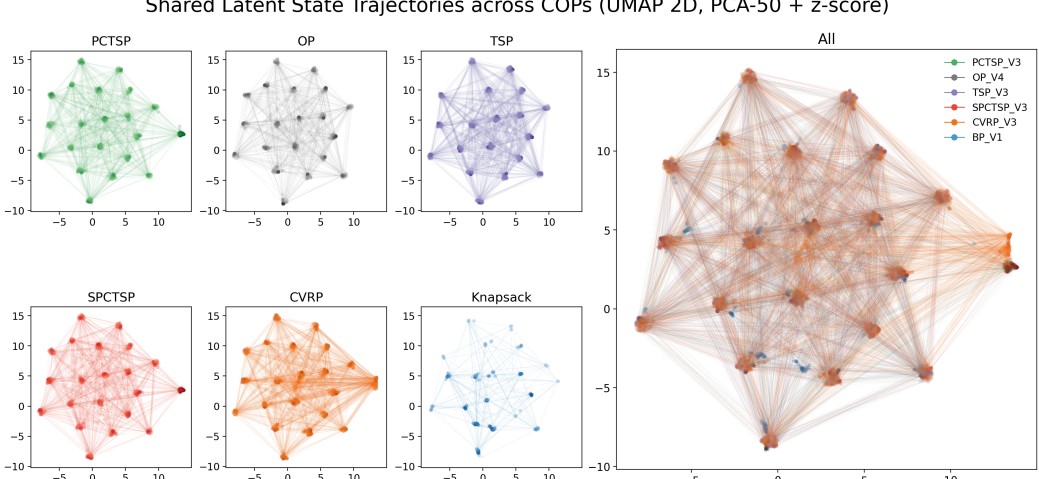

Figure 10: Visualization of trajectory representations across COPs at $N = 20$: for each COP, each point denotes the last-layer hidden state at the action-token position along expert rollouts, projected to 2D via PCA-50 followed by UMAP.

# E ANALYSIS OF CROSS-PROBLEM REPRESENTATIONS

## E.1 TRAJECTORY VISUALIZATION ACROSS HETEROGENEOUS COPS

To analyze whether COFormer learns shared representations across heterogeneous COPs, we visualize the hidden trajectories induced by the $N = 20$ multi-COP checkpoint jointly trained on the eight problems listed in Table 1. For each COP, we randomly sample 100 expert trajectories from the corresponding dataset and feed their tokenized full MDP trajectories into the model, running a forward pass without parameter updates. At every decision step $t$ along these expert episodes, we take the hidden state vector at the action-token position (i.e., the vector used to predict $a_t$) of the last Transformer block as the representation of the current MDP state $s_t$. We then standardize these vectors, reduce their dimensionality to 50 via PCA, and project them into a 2D space using UMAP. The PCA and UMAP mappings are fitted jointly on the union of trajectories so that states from different COPs share a common embedding space.

Due to the limited context length $L = 1000$ of this checkpoint, we restrict the visualization to problems whose complete expert trajectories can be reliably processed (the five routing problems and Knapsack). For FFSP and 3DBP, full rollouts typically exceed this length; using truncated subsequences changes the state distribution (e.g., missing early stages of the scheduling or packing process) and empirically leads to substantially noisier and less interpretable projections. To avoid over-interpreting such artifacts, we do not include FFSP and 3DBP in the trajectory visualization and leave a more systematic analysis of their representations with longer-context models to future work.

As shown in Figure 10, each point corresponds to an MDP state, and the edges connect consecutive states along a trajectory. We observe that trajectories from different problems occupy highly overlapping manifolds in the projected space, and the MDP states across heterogeneous COPs are only weakly separated. In particular, many intermediate Knapsack states lie in the interior of the routing trajectories. This pattern suggests that the model organizes states primarily according to high-level semantics such as resource-usage patterns, rather than by explicit problem identity, field order, or task-specific positional regularities, which is consistent with the few-shot and zero-shot experiments in Section 4.4 where COFormer trained on one subset of routing problems can be efficiently adapted to other routing problems with limited additional data.

## E.2 FEW-SHOT TRANSFER ACROSS HETEROGENEOUS COPs

Motivated by the overlap between Knapsack and routing trajectories observed in Figure 10, we further test whether the shared latent structure can be exploited for cross-family transfer. Starting from the multi-routing checkpoint trained on TSP, CVRP, OP, PCTSP, and SPCTSP, we fine-tune COFormer on Knapsack instances and compare it to a model trained on Knapsack from scratch. We use the same model architecture as in Table 1. The fine-tuning learning rate is fixed at $1e-4$, and each epoch uses 2400 training instances. This fine-tuning takes about 9 minutes for 15 epochs on 2 NVIDIA A100 GPUs.

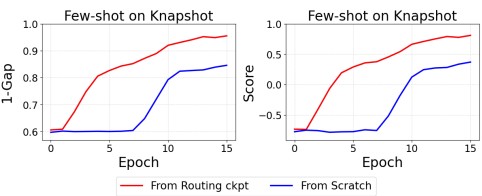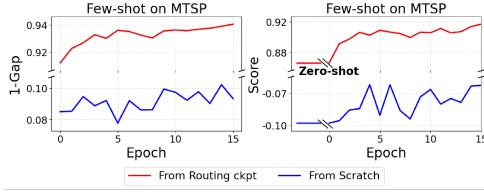

(a) Few-shot fine-tuning on Knapsack      (b) Few-shot fine-tuning on Modified-TSP

Figure 11: Few-shot adaptation of COFormer to heterogeneous COPs and problems with unseen constraints: fine-tuning from the routing-pretrained checkpoint can leverage shared semantic representations to achieve strong few-shot performance on Knapsack and even strong zero-shot performance on Modified-TSP, outperforming training from scratch.

As shown in Figure 11(a), the routing-initialized mode consistently achieves better performance than the model trained from scratch on both metrics. Although the two models start from a similar performance level, the routing-initialized one improves much faster, surpasses the best from-scratch performance after only 2 epochs, and converges to a clearly higher final performance (e.g., 1-Gap around $95\%$ vs. $84\%$) under limited data budget. Together with the trajectory-level overlap between Knapsack and routing, this result suggests that parameters learned on routing problems capture representations that transfer effectively to Knapsack, despite differences in state structure and combinatorial constraints.

## E.3 ZERO-SHOT AND FEW-SHOT TRANSFER TO PROBLEMS WITH UNSEEN CONSTRAINTS

We further investigate whether COFormer can generalize to problems with unseen constraint types. We construct a Modified TSP (denoted as MTSP) that augments the standard TSP with a minimum tour-length constraint and relaxes the visit-once constraint: the objective is to find a tour that starts and ends at the same city, visits all cities (allowing revisits), and is as short as possible while satisfying the minimum-length constraint (see Sec. A.9 for details). Starting from the same routing-pretrained checkpoint as Sec. E.2, we fine-tune COFormer on MTSP instances and compare it to a model trained on MTSP from scratch. The learning rate is fixed at $1e-4$, using 9,000 training instances in total, and it takes around 10 minutes for 15 epochs on 2 NVIDIA A100 GPUs.

As shown in Figure 11(b), the routing-pretrained model already achieves strong zero-shot performance on MTSP (about $91\%$ 1-Gap and $86\%$ Score) and continues to improve during fine-tuning, whereas the from-scratch model shows almost no progress under the same data and compute budget. These results indicate that, at least on the COPs considered here, COFormer exhibits non-trivial cross-constraint transfer ability, leveraging representations learned on standard routing problems to solve problems with substantially different constraints.

## F ABLATIONS ON NUMERIC TOKENIZATION

In all experiments, we construct input sequences using the tokenization scheme described in Appendix B. All continuous features are encoded by a two-step procedure: (i) task-agnostic normalization of raw values into a bounded range via a $\mu$-law transformation, and (ii) discretization into a fixed number of bins, which are then mapped to integer token IDs. This design unifies heterogeneous COPs with very different numerical scales (e.g., distances, prizes, penalties) into a single

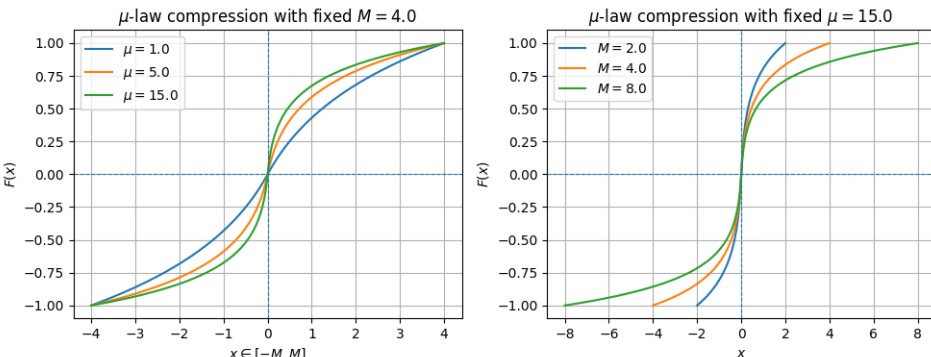

Figure 12: Illustration of the $\mu$-law transformation used for numeric tokenization. Left: effect of varying $\mu$ with fixed $M = 4.0$; larger $\mu$ increases nonlinearity and allocates higher resolution to small magnitudes. Right: effect of varying $M$ with fixed $\mu = 15.0$; larger $M$ expands the covered value range.

numeric token space, and avoids introducing problem-specific encoders (such as per-task MLPs or GNNs) that would add inductive bias and reduce the generality of the shared Transformer backbone.

As discussed in Appendix A, all continuous fields in our benchmark COPs lie in $[0.0, 4.0]$ (the largest value is the `length` field in OP100). Therefore, our default configuration applies a $\mu$-law transformation with range parameter $M = 4.0$ to map these values into $[0.0, 1.0]$, and then discretizes this interval into 1,800 bins. As shown in Figure 12, the $\mu$-law transformation is a monotone compression that preserves value ordering while reallocating resolution along the range. The behavior of this tokenization scheme is governed by three hyperparameters:

- **Bin count.** Increasing the number of bins reduces quantization error, but also enlarges the numeric vocabulary and creates more rare tokens whose embeddings are harder to train.

- **Range parameter $M$.** Controls the overall compression range. To maximize effective discretization resolution for a fixed number of bins, it should be chosen to just cover the global range of all continuous fields and map it tightly into the interval to be discretized. As shown in the right panel of Figure 12, larger $M$ corresponds to a horizontally stretched curve.

- **Nonlinearity parameter $\mu$.** Controls the degree of nonlinearity in compression: larger $\mu$ allocates higher resolution to small magnitudes at the cost of coarser resolution for larger values. This is advantageous for small quantities such as prizes and penalties, but can potentially distort approximately uniform quantities such as 2D coordinates if the nonlinearity is too strong. The left panel of Figure 12 shows how increasing $\mu$ makes the curve more concentrated around the origin.

To assess the sensitivity of COFormer to these design choices, we conduct an ablation on six routing problems of size $N = 20$: TSP, CVRP, OP, PCTSP, SPCTSP, and ATSP. We use the same model architecture as in Table 1, follow the training recipe for *COFormer-direct*, and consider the following configurations:

- **v1 (1,800 bins).** Default $\mu$-law setting described above on $[0, 4]$, and discretizes the resulting range $[0.0, 1.0]$ into 1,800 bins.

- **v2 (1,800–6,800 bins).** A more linear, symmetric variant that shifts continuous values so that their ranges are approximately symmetric around the origin (e.g., coordinates in $[-0.5, 0.5]$, OP100 `length` in $[-2.0, 2.0]$), applies $\mu$-law with $M = 2.0$ and $\mu = 15.0$, and discretizes the resulting range $[-1.0, 1.0]$ into 1,800, 2,800, 4,800, or 6,800 bins.

For v2 configurations with more bins (2,800, 4,800, and 6,800), we apply a light local smoothing of numeric embeddings to mitigate sparsity: at training time, the embedding of a numeric token is computed as a truncated Gaussian-weighted average of embeddings in a small neighborhood (50 adjacent bins), with the variance annealed over training.

Table 9: Ablations on $\mu$-law variants and discretization resolution on six routing COPs.

| Setting | TSP | CVRP | OP | PCTSP | SPCTSP | ATSP |
|---|---|---|---|---|---|---|
| v1 (1,800) | 99.13% | **91.35%** | 88.78% | 98.11% | **100.72%** | 92.58% |
| v2 (1,800) | 99.25% | 88.79% | 89.13% | 98.21% | 100.60% | 93.67% |
| v2 (2,800) | **99.26%** | 89.73% | **89.88%** | 98.16% | 100.53% | **94.89%** |
| v2 (4,800) | 99.17% | 89.40% | 88.81% | **98.30%** | **100.72%** | 92.55% |
| v2 (6,800) | 99.23% | 88.93% | 88.06% | 98.19% | 100.61% | 93.65% |

Table 9 reports the score metric for all configurations across the six COPs. We observe that training remains stable across all settings, and the performance differences are modest: relative gaps are typically within 1–2% on each task, and there is no consistent monotonic improvement beyond 1,800 bins. For example, on TSP and PCTSP all configurations remain within 0.2–0.3% of each other, while on the other routing tasks the best and worst configurations differ by only a few percent, without changing the qualitative conclusions of the main results.

Overall, these ablations indicate that COFormer is not overly sensitive to the exact choice of $\mu$-law variant or discretization resolution within a broad range of reasonable settings. The numeric tokenization scheme is used primarily to bring heterogeneous continuous features into a shared discrete space with adequate resolution around small values; within this regime, COFormer's performance remains robust, and the main conclusions of the paper do not rely on a finely tuned choice of scaling or bin count.

## G    EVALUATION DETAILS AND TRAINING PROCESS REPORTS.

We provide more details of the implementation for reproducibility. The hyperparameter used and the environment settings for Table 1, Table 3 and Table 4 are illustrated in Table 8. The detailed training process in terms of loss is shown in Figure 13. GATO/DB1 converges much slower than our model across all tasks.

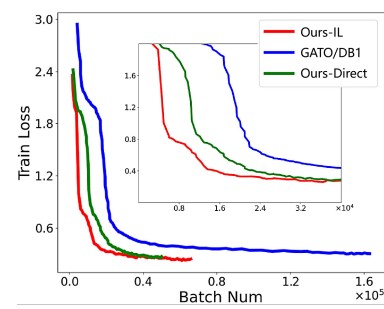

Figure 13: The loss curves during training.

## H    LLM USAGE STATEMENT

We used LLM only for minor language polishing and grammar checking of the manuscript. The LLM did not contribute to the research ideas, methodology, data analysis, or interpretation of the results.

