# OpenReview forum: "COFormer: Towards a Foundation Model for Solving Combinatorial Optimization Problems"
_ICLR.cc/2026/Conference — Submitted to ICLR 2026_

### Official Review · Reviewer_wAjh · 2025-10-20

**Soundness:** 2
**Presentation:** 2
**Contribution:** 3
**Rating:** 4
**Confidence:** 4

**Summary:**

This paper introduces COFormer, a Transformer-based framework designed to solve diverse combinatorial optimization problems (COPs) within a single unified architecture. Drawing inspiration from next-token prediction paradigms (as in large language models), COFormer reformulates COPs as sequential decision processes, representing their trajectories as token sequences. The framework introduces two key ideas:
- A CO-prefix mechanism that encodes static problem information to reduce sequence length.
- A three-stage learning scheme combining imitation learning (forward dynamics and policy generation) with reinforcement learning fine-tuning.

The authors evaluate COFormer on eight distinct COPs (e.g., TSP, VRP, Knapsack, FFSP, 3D Bin Packing), showing strong multi-problem generalization, few-shot transfer, and even some zero-shot capabilities.

**Strengths:**

**Originality**

- The paper addresses a key open challenge in neural combinatorial optimization by developing a unified, cross-problem model. This aligns well with the current research trajectory aiming for foundation models for decision-making.
- The use of a CO-prefix to separate static and dynamic features is elegant and practically motivated, reducing token length and improving efficiency.

**Significance**

- The empirical evaluation spans eight distinct CO problems, including both graph-based (TSP, VRP) and non-graph-based (FFSP, 3DBP) tasks. Considerably broader than most prior “generalist” NCO works.
- The reported results show strong generalization and competitive performance against specialized baselines.

**Clarity & Quality**
- The paper communicates its motivation, contributions, and architecture clearly, with well-structured exposition and helpful figures, though technical descriptions, especially regarding tokenization, could be improved.
- Quality (Strength): The learning process is well motivated and comprehensive

**Weaknesses:**

**Lacking Clarity**

The paper lacks a clear and rigorous description of how heterogeneous problem features are represented in the token space. The paper’s description of the tokenization process (Sec. 3.1.2, App. B.1) lacks sufficient clarity to reproduce or fully understand how heterogeneous problem states are unified.

While it states that both continuous and discrete values are flattened and mapped to integer token IDs, it is unclear how structural or semantic distinctions (e.g., between coordinates, demands, processing times) are preserved.

The brief mention of “arbitrary key–value dictionaries” (App. B.1) implies flexible data interfaces but does not specify whether keys are tokenized, whether per-problem schemas are used, or how feature ordering is standardized.

As a result, it is difficult to assess whether the model genuinely learns a shared latent space across problem types, or whether it relies on per-problem positional regularities hidden in the data pipeline.

This opacity is problematic for reproducibility and for understanding how COFormer achieves semantic generalization.


**Lack of clarity and analysis on semantic representation learning**

The paper does not clearly explain how COFormer achieves true semantic unification across heterogeneous COPs and would benefit from a deeper analysis of why COFormer can generalize across problems. For instance, are representations for routing and scheduling problems aligned in the same latent space? How sensitive is performance to feature ordering or scaling in the CO-prefix?

If features are distinguished mainly by positional or problem-specific ordering rather than shared meaning, the model’s generality may be superficial.

No ablations or visualization of learned embeddings are provided to demonstrate that the learned embeddings capture transferable structure.


**Ablation analysis missing for representation choices**

No experiments isolate the effects of specific design choices (e.g., with/without µ-law scaling, or alternative encoding of continuous features). It’s unclear how sensitive performance is to these details

**Baseline selection in experimental results**

In the experimental section, results of more recent NCO models (e.g. [1], [2], [3], [4], just to name a few) should be included. Also, comparisons with specialist models seem incomplete, as COFormer often uses sampling while specialists use greedy generation mode. For FFSP for example, MatNet achieves an average makespan of 25.4 on (probably the same) FFSP test instances with sampling enabled. [4] even brings this down to 24.9, giving a performance gap of 14%, which appears very large for such small instances.

**Missing training times**

The paper claims to improve training efficiency, but does not report wall-clock training times.

-----
[1] Grinsztajn, N., Furelos-Blanco, D., Surana, S., Bonnet, C., & Barrett, T. (2023). Winner takes it all: Training performant rl populations for combinatorial optimization. Advances in Neural Information Processing Systems, 36, 48485-48509.

[2] Liao, Z., Chen, J., Wang, D., Zhang, Z. &amp; Wang, J.. (2025). BOPO: Neural Combinatorial Optimization via Best-anchored and Objective-guided Preference Optimization. Proceedings of the 42nd International Conference on Machine Learning

[3] Pirnay, J., & Grimm, D. G. (2024). Self-improvement for neural combinatorial optimization: Sample without replacement, but improvement. arXiv preprint arXiv:2403.15180.

[4] Hottung, A., Mahajan, M., & Tierney, K. (2024). PolyNet: Learning diverse solution strategies for neural combinatorial optimization. arXiv preprint arXiv:2402.14048.

**Questions:**

- How exactly is the feature ordering determined when flattening MDP states across problem types? Are keys (feature names) tokenized or embedded, or are they implicit via positional order? Can you provide a concrete example of a serialized sequence (prefix + trajectory) for two distinct problems (e.g., VRP and FFSP)?

- How does the model distinguish between, e.g., a coordinate token and a demand token, if both occupy the same token ID range? How does the model, which problem it is currently solving?

- Did you observe shared latent structure between different problems (e.g., via attention visualization or representation similarity)?

- Can the model generalize to problems with different constraint types not seen during training?

---

> ### Author Response · Authors · 2025-12-03
>
> **[W1/Q1]**：
>
> We thank the reviewer for pointing out the ambiguity in our description of the tokenization pipeline. We follow the flattening scheme used in GATO: for both the CO-prefix and the state, we first sort keys in lexicographic order and then concatenate and flatten all value tensors. In this way, **keys are encoded implicitly via their position in a fixed schema**, rather than through separate key tokens. The per-problem key orders are summarized below (rows are ordered lexicographically by key name):
>
> - CO-prefix key order
>
>   | Knapshot     | FFSP      | 3DBP | TSP      | CVRP      | OP        | PCTSP     | SPCTSP    |
>   | ------------ | --------- | ---- | -------- | --------- | --------- | --------- | --------- |
>   | item_values  | durations | -    | position | demand    | pos_depot | penalty   | det_prize |
>   | item_volumes |           |      |          | pos_depot | pos_node  | pos_depot | penalty   |
>   |              |           |      |          | pos_node  | prize     | pos_node  | pos_depot |
>   |              |           |      |          |           |           | prize     | pos_node  |
>
> - State key order
>
>   | Knapshot      | FFSP          | 3DBP          | TSP              | CVRP             | OP               | PCTSP            | SPCTSP           |
>   | ------------- | ------------- | ------------- | ---------------- | ---------------- | ---------------- | ---------------- | ---------------- |
>   | capacity_left | machine_query | current_state | current_position | capacity         | current_position | current_position | current_position |
>   |               |               |               |                  | current_position | length           | prize2go         | stoc_prize2go    |
>   |               |               |               |                  |                  | prize            |                  |                  |
>
> Within each problem, the ordering of values is induced purely by the **lexicographic sorting of human-readable key names** and the natural structure of the state, without any manual engineering of key names to enforce a particular cross-problem ordering. Take the CO-Prefix of FFSP and CVRP as an example:
>
> - FFSP has $N\times M$ `durations` tokens in the CO-Prefix.
> - CVRP has $N$ `demand` tokens followed by 2 `pos_depot` tokens and then $2N$ ` pos_node` tokens in the CO-Prefix.
>
> It is important to Note that we deliberately chose **not** to introduce explicit key tokens (e.g., assigning a special token ID to each field type such as 1 for `demand`, 2 for `position`) in order to keep the interface as generic as possible and avoid hard-coding problem-specific inductive biases that could hinder generalization to COPs with unseen semantic fields. Instead, all scalar features are mapped into a **shared numeric token space** using task-agnostic normalization and μ-law discretization. As a result, semantically similar fields tend to share similar distributions (e.g., coordinates such as `pos_node` and `current_position` follow approximately uniform distributions; “smaller `capacity`” and “smaller `prize`” are consistently mapped to smaller token IDs), which in turn encourages **shared semantic embeddings** in the model.

---

> ### Author Response · Authors · 2025-12-03
>
> **[Q2]**
>
> For **problem-type semantics**, each trajectory is always preceded by a CO-prefix that encodes the full static instance, and the state contains COP-specific dynamic fields. Different COPs induce **systematically different prefix/state patterns in token length and statistics** (e.g., FFSP has an $N \times M$ `durations` matrix in the prefix, whereas Knapsack has item-wise `item_values`/`item_volumes`). Because instances from all eight COPs are interleaved during training, the Transformer can infer which problem it is solving from these prefix+state patterns, without requiring an explicit problem-ID token. This implicit inference mechanism performs well in our experiments.
>
> For **feature-type semantics**, numeric token IDs are shared but used in different **contexts and value distributions**. Coordinate-like fields (e.g., `pos_node`, `current_position`) are approximately uniform over the unit square, whereas demand-/prize-like fields appear as single scalars tied to nodes and directly affect feasibility and objective terms (e.g., capacity updates in CVRP and accumulated prize in PCTSP). Self-attention over this structured context, together with IL/RL training, allows COFormer to assign different effective hidden states to “coordinate-like” and “demand-like” usages even though they occupy the same numeric token range.

---

> ### Author Response · Authors · 2025-12-03
>
> **[W2/Q3/Q4]**
>
> To address the reviewer’s request for a deeper analysis of **semantic unification**, **shared latent structure**, and **generalization to new constraint types** across heterogeneous COPs, we add new experiments and visualizations in Appendix E that explicitly examine whether COFormer learns a **shared latent space** and **transferable representations**:
>
> - **Visualization of learned representations (Figure 10).** UMAP embeddings of latent states for five routing problems and Knapsack show that their trajectories occupy a largely overlapping manifold in the learned representation space, providing direct evidence of shared latent structure across different COP families.
> - **Cross-family transfer (Figure 11(a)).** Starting from a multi-routing checkpoint, fine-tuning on Knapsack leads to faster convergence and better final performance than training a Knapsack-only model from scratch. This shows that representations learned on routing COPs can be effectively reused for a different family (packing), and offers a concrete explanation of **why COFormer can generalize across problems**.
> - **Within-family transfer and new constraints (Figures 4 and 11(b)).** Within routing, we study transfer across different routing COPs and across **different constraint types**. We observe strong few-shot improvements and non-trivial zero-shot performance on TSP variants whose constraints were not present in the pre-training mix, indicating that COFormer learns a common routing representation that can be adapted to new constraint patterns with limited additional data. This shows that COFormer can generalize to problems with new constraint types not seen during training, particularly within the routing family.
>
> These results jointly demonstrate that the learned embeddings capture **transferable structure** and that COFormer’s generalization is not merely superficial or driven only by problem-specific ordering.
>
> For **FFSP and 3DBP**, under our present setup we do not consistently observe clear improvements over training these tasks from scratch. Our preliminary analysis suggests that these problems induce much longer MDP trajectories; with the current context-length limit, rollouts must be split into shorter subsequences whose distribution differs from full trajectories, which can make it harder for the model to fully exploit cross-task representation sharing. Importantly, however, FFSP and 3DBP are still handled by the *same* shared Transformer backbone in the multi-task setting (Table 1); the limitation lies in the strength of few-shot transfer under constrained context length, not in an inability to incorporate these problems into a unified architecture. We explicitly acknowledge this limitation in Appendix E and view longer-context architectures and trajectory-aware segmentation as promising directions for further improving cross-family sharing.
>
> Regarding **sensitivity to feature ordering and scaling in the CO-prefix**, all COPs share a **single numeric tokenization scheme** with task-agnostic normalization and μ-law discretization, yet we still observe strong within-family and cross-family transfer as above. This suggests that COFormer is not simply exploiting idiosyncratic feature scales. Moreover, as discussed in our reply to W1/Q1, the prefix/state schemas for different COPs differ substantially in length and structure, and keys are ordered only via a simple lexicographic rule on human-readable names, without manually engineering cross-problem alignments. The fact that robust transfer emerges under such heterogeneous schemas indicates that COFormer is learning **semantically meaningful shared structure** rather than relying on a finely tuned hand-crafted ordering.

---

> ### Author Response · Authors · 2025-12-03
>
> **[W3]**
>
> In COFormer, all continuous features are embedded via a unified $\mu$-law + binning scheme: raw values (with different scales across COPs) are first normalized into a fixed range by a $\mu$-law transformation, and then discretized into a fixed number of bins which are mapped to integer token IDs. This allows us to represent heterogeneous continuous fields in a single numeric token space. We **deliberately avoid** per-problem MLP/GNN encoders in order not to introduce additional problem-specific inductive bias into the shared Transformer backbone.
>
> In the revised version, we add Appendix F to explicitly study the trade-off between discretization resolution and stability. On six routing problems with $N=20$ (TSP, CVRP, OP, PCTSP, SPCTSP, ATSP), using the same architecture and training recipe as *COFormer-direct*, we vary both the $\mu$-law variant and the bin count (from 1,800 up to 6,800 bins). Across all settings, training remains stable and performance differences are modest (typically within 1–2% on each task), with no consistent monotone gain beyond default setting. These results indicate that COFormer is not overly sensitive to the specific choice of $\mu$-law parameters or bin count, and that our main conclusions do not rely on a finely tuned trade-off between discretization resolution and quantization noise.

---

> ### Author Response · Authors · 2025-12-03
>
> **[W4]**
>
> We thank the reviewer for pointing out these recent NCO methods. We fully agree that the NCO landscape is evolving rapidly and that these works are highly relevant. Our experimental design aims to balance **diversity of baselines** with **computational feasibility**: in the current version we already include both specialist models (e.g., PCT for 3DBP, POMO for TSP and Knapsack) and a recent generalist baseline (GOAL, 2025). Training and evaluating these baselines across up to eight COPs is already computationally demanding, and other reviewers have explicitly noted the breadth of our evaluation — for example, PG5P comments that *“the cross-problem evaluation is extensive; the benchmark setup itself constitutes a valuable contribution to the community”*, and wAjh highlights that *“the empirical evaluation spans eight distinct CO problems … considerably broader than most prior ‘generalist’ NCO works.”*
>
> During the rebuttal period, our compute resources were primarily allocated to additional experiments requested by reviewers (e.g., CVRP1000, new cross-problem transfer and representation analyses), so we were unfortunately not able to re-implement these specialized methods under our setup. We will add a discussion of these methods in the related-work section and explicitly state that incorporating them into our benchmark is an important direction for future work.
>
> Regarding FFSP in particular, we appreciate the reviewer’s detailed comparison and would like to clarify two points to better contextualize the reported gaps:
>
> 1. The FFSP numbers cited for MatNet and PolyNet (e.g., 25.4 and 24.9) are obtained under a **×128 sampling/augmentation** setting, whereas in our current experiments COFormer is evaluated with at most **×16 sampling** (line 364-365). Thus, the raw objective values are not directly comparable.
> 2. Our FFSP training data are generated by **MatNet with greedy decoding**, and on our FFSP test set this expert model achieves an average makespan of about **27.3**. Under pure supervised learning, it is therefore expected that the *COFormer direct* and *COFormer IL* will obtain test objectives slightly above this teacher level. In this context, it is reasonable that our reported COFormer performance does not match the 24.9 makespan reported for MatNet/PolyNet under much stronger ×128 sampling settings. In this work we treat FFSP as one of eight tasks within a single unified training pipeline, rather than as a standalone SOTA target.
>
> It is important to emphasize that our goal in this paper is not to claim SOTA performance on FFSP (or any single COP) in isolation, but to show that a single COFormer model can handle routing, packing, scheduling, and 3D bin packing within **one unified architecture** while still being **reasonably competitive** with established baselines, and to demonstrate its zero-shot and few-shot generalization capabilities across COPs. These latter aspects are extensively evaluated in the main text and appendices, and we will clarify this positioning more explicitly in the revised manuscript.

---

> ### Author Response · Authors · 2025-12-03
>
> **[W5]**
>
> Our training-efficiency claim is made with respect to the **supervised learning stage**. For the main experiments in Table 1, run on the same hardware and under the same training setup (with detailed hyperparameters given in Table 8), the GATO/DB1 baseline requires about **77 hours** of supervised training, whereas *COFormer-direct* takes about **30 hours**. Thus, for the supervised stage that is directly comparable across methods, COFormer reduces wall-clock training time by up to roughly **2.5×** relative to the GATO/DB1 baseline.
>
> Even if we further include the additional **dynamics forward stage** pretraining, *COFormer-IL* still needs only about **56 hours** in total, which is still roughly **30% faster** than the GATO/DB1 baseline. Note that RL fine-tuning is applied only to COFormer and has no direct GATO/DB1 counterpart, so its additional cost is not included in this supervised-efficiency comparison.
>
> The main source of this gain is the **CO-prefix** mechanism: by compressing all static problem information into a short prefix, each trajectory sequence becomes significantly shorter, so that for a fixed context length a much larger fraction of tokens are **action tokens** on which the loss is computed. This effectively increases the number of decisions processed per update for comparable compute, leading to faster supervised training in wall-clock time.

---

### Official Review · Reviewer_PG5P · 2025-10-21

**Soundness:** 4
**Presentation:** 4
**Contribution:** 3
**Rating:** 8
**Confidence:** 4

**Summary:**

This paper proposes **COFormer**, a unified transformer-based framework designed to solve diverse combinatorial optimization problems (COPs) using a single model and shared parameters. Inspired by next-token prediction in large language models, the authors formulate COP solving as a sequential decision process, tokenizing both states and actions into unified trajectories. A **CO-prefix** mechanism is introduced to compactly encode static problem information, and a **three-stage learning scheme** (dynamics imitation, policy imitation, and reinforcement fine-tuning) is proposed to improve training efficiency and generalization. Evaluated on eight distinct COPs (e.g., TSP, VRP, Knapsack, FFSP, 3DBP), COFormer matches or surpasses problem-specific NCO methods and prior generalist baselines, while demonstrating strong few-shot and even zero-shot generalization to unseen problem types.

**Strengths:**

- **\[S1]** The idea of building a _foundation model_ for combinatorial optimization via tokenization is interesting and, to the best of my knowledge, novel.

- **\[S2]** The **CO-prefix** mechanism is a clever and effective way to handle static problem information efficiently.

- **\[S3]** The proposed **multi-stage training pipeline** is clearly structured and modular, improving interpretability and training stability.

- **\[S4]** The **cross-problem evaluation** is extensive; the benchmark setup itself constitutes a valuable contribution to the community.

- **\[S5]** The model demonstrates **promising generalization** abilities, including few-shot and even zero-shot adaptation.

- **\[S6]** The approach exhibits **good scalability**, effectively handling large instances such as TSP1000, supporting its claim as a potential foundation model for COPs.

**Weaknesses:**

See “Questions” below.

**Questions:**

- **\[Q1]** In COFormer, both states and actions are tokenized and embedded into the same latent space, with their roles distinguished only by positional and separator tokens. While this enables a unified architecture across heterogeneous COPs, it also “blurs” the structural distinction between state and action representations that is central to the underlying MDP formulation. Could the authors elaborate on how this design choice affects learning efficiency and generalization? Specifically, have they compared this unified tokenization scheme to architectures where states and actions are modeled in separate embedding spaces (e.g., distinct encoders or dual-stream attention), which might preserve causal structure and reduce representational ambiguity?

- **\[Q2]** Many combinatorial optimization problems possess strong structural priors—for example, graphs in routing tasks or bipartite relations in scheduling—that can be compactly and efficiently represented in their native forms. In COFormer, however, these structured instances are flattened into token sequences, which may obscure their relational topology and lead to substantial increases in sequence length. Could the authors discuss how this loss of structural inductive bias impacts model scalability and sample efficiency? Have they considered incorporating lightweight structure-aware modules (e.g., graph or set encoders) that could preserve structural efficiency while maintaining the unified token-based framework?

- **\[Q3]** The three-stage training pipeline (dynamics imitation → policy imitation → RL fine-tuning) is a key design choice. Could the authors elaborate on how critical each stage is to final performance? For example, what happens if the dynamics stage is skipped, or if RL fine-tuning is applied directly after policy imitation? An ablation or sensitivity analysis would help clarify the necessity of this modular training design.

- **\[Q4]** Continuous features are discretized via μ-law transformation into 1,800 bins. Could the authors discuss the trade-off between discretization resolution and training stability? In particular, how sensitive is performance to the bin count or to the quantization noise introduced by this process?

---

> ### Comment · Reviewer_PG5P · 2025-11-27
>
> As the authors did not reply to the other reviewers, and I think the other reviewers raised multiple fair concerns. I will not champion for this submission.

---

> > ### Author Response · Authors · 2025-11-27
> >
> > **We sincerely apologize for the delayed response.**
> >
> > Due to limited compute resources, the additional experiments suggested by the reviewers took longer to finish than expected, and we were not able to respond promptly. This is fully our responsibility, and we apologize for not replying sooner.
> >
> > We have now uploaded an updated PDF that includes:
> >
> > - **Appendix E.1:** new analyses of **semantic representations across heterogeneous COPs**.
> > - **Appendix E.2–E.3:** additional **cross-COP** and **cross-constraint few-shot transfer** experiments showing that COFormer indeed learns shared representations.
> > - **Table 5:** added **CVRP1000** results, further demonstrating that our training paradigm can scale effectively to larger instances.
> >
> > We are also running the **ATSP** experiments mentioned by the reviewers, and we will provide a **more detailed, point-by-point reply** as soon as all experiments are finished.
> >
> > We sincerely appreciate the reviewers’ constructive comments and will continue addressing all points carefully.

---

> > > ### Comment · Reviewer_PG5P · 2025-11-28
> > >
> > > Thanks for the clarification. I (and the other reviewers) will be waiting for your rebuttal.

---

> ### Author Response · Authors · 2025-12-03
>
> **[Q1]**
>
> We thank the reviewer for this insightful question. From a policy-learning perspective, the core objective is to map **states and the trajectory history** to the next action. Consequently, most representational capacity is naturally devoted to encoding CO-prefix and state tokens, while action tokens mainly serve as **index selectors** (e.g., “choose city *i*” or “choose item *j*”) whose semantic meaning is defined by the surrounding state context rather than by their standalone embeddings. In each trajectory, CO-prefix and state tokens dominate in number, and the policy loss is computed from the encoded state and trajectory history. As a result, the gradients that shape the shared latent space are largely driven by learning good state and trajectory representations, and the relatively few action tokens do not significantly distort these representations.
>
> Importantly, the MDP structure is not encoded solely through the embedding table, but through the **sequence pattern, attention masks, and training losses**. Concretely, we use the following mechanisms to keep state and action roles clearly separated, even though they share the same embedding matrix:
>
> - **Explicit sequence pattern:** We introduce dedicated prefix-splitter and action-splitter tokens to define a clear state–action alternation in the trajectory.
> - **Role-specific training signals:** The dynamics loss (in stage 1) is applied only on state positions, while the policy loss (in stage 2) is applied only on action positions, enforcing a functional distinction between state prediction and action selection.
> - **Hybrid attention for static vs. dynamic tokens:** Static problem information in the CO-prefix is processed with bidirectional self-attention, allowing all static tokens to interact freely and form a compact global summary of the instance. In contrast, the dynamic state–action part uses causal attention. This hybrid scheme exploits static structure while preserving the causal constraints of the underlying MDP.
>
> Regarding learning efficiency and generalization, a unified token space has two practical benefits. First, it improves **parameter sharing**: the same embedding matrix is used to encode entities that appear both as part of the state and as targets of actions, which empirically leads to stable optimization without extra architectural overhead. Second, by avoiding separate state/action encoders or dual-stream attention, we reduce model complexity and simplify training, making it easier to scale to larger instances and to heterogeneous COPs under a fixed compute budget. Empirically, the unified design already leads to strong cross-problem generalization: Appendix E shows that trajectories from different COPs occupy overlapping manifolds in the latent space, and our few-shot/zero-shot experiments in Figure 4, Figure 9, and Figure 11 demonstrate that the learned representations transfer well across tasks.
>
> Exploring role-specific embeddings or dual-stream architectures is an interesting direction for potentially further improving sample efficiency, and we plan to investigate this in future work.

---

> ### Author Response · Authors · 2025-12-03
>
> **[Q2]**
>
> We appreciate the reviewer’s thoughtful question about structural priors. Our core design principle is to **tokenize away problem-specific differences**: all COPs are mapped into unified token trajectories so that their heterogeneity is reflected primarily in **sequence length and content**, and a **single Transformer** can learn shared structure across tasks. To maximize this cross-problem generality, we intentionally do **not** introduce any problem-specific, structure-aware encoders, as also discussed in Appendix B.1 (lines 946–956). This keeps COFormer as a fully token-based backbone that can, in principle, be extended to new COPs without redesigning dedicated encoders.
>
> We agree that removing explicit structural inductive bias can lead to longer token sequences and potentially lower sample efficiency. **COFormer mitigates this efficiency loss through the CO-prefix design**, which aggressively compresses static problem information into a short prefix so that, for a fixed context length, a much larger fraction of tokens are action tokens on which the loss is computed under teacher-forcing-style training. In other words, although we give up structure-aware encoders for the sake of generality, CO-prefix increases the number of decisions processed per update and thereby compensates, to a large extent, for the efficiency that would otherwise be provided by such encoders. Concretely, for the main experiments in Table 1, in the policy learning stage (Stage 2) directly comparable to GATO/DB1, *COFormer-direct* trains in about **30 hours** versus **77 hours** for GATO/DB1 (≈2.5× faster), and even including the additional dynamics-forward pretraining, *COFormer-IL* needs only about **56 hours**, still ≈30% faster. As a result of CO-prefix compression, even for large instances such as **TSP1000** and **CVRP1000**, the total sequence length remains in the **6k–8.5k** range—orders of magnitude shorter than directly tokenizing raw MDP trajectories as in GATO. The large-scale results in **Table 5** show that under this compressed representation, COFormer remains computationally scalable and RL fine-tuning still yields meaningful improvements over imitation learning.
>
> Looking ahead, we agree that incorporating **lightweight structure-aware encoders** is a promising extension. A natural way to integrate them while preserving the unified token interface is to let certain CO-prefix or state key–value fields be embedded by a **COP-family–specific encoder** (e.g., for routing or scheduling), which keeps COFormer’s token-based backbone unchanged while injecting family-level structural priors. In preliminary experiments, for example, we introduced a small MLP encoder for 2D coordinates in routing tasks, which replaced raw coordinate tokens, roughly halved the coordinate-related token sequence length, and led to faster convergence and improved data efficiency. However, we ultimately decided to omit such family-specific encoders in the current paper in order to maximize architectural uniformity across all COPs. We plan to further explore these structure-aware variants in future work, especially designs that reuse a single encoder within a COP family to better balance generality and sample efficiency.

---

> ### Author Response · Authors · 2025-12-03
>
> **[Q3]**
>
> We thank the reviewer for this question. The three-stage pipeline is designed so that each stage plays a distinct role:
>
> - **Stage 1 (Dynamics imitation)** learns an explicit dynamics model, which enhance the model’s awareness of action consequences and provides a good initialization of state representations.
> - **Stage 2 (Policy imitation)** brings the policy close to expert performance under supervised learning.
> - **Stage 3 (RL fine-tuning)** then improves beyond imitation by directly optimizing the task reward.
>
> In the main experiments, an ablation over the three configurations is **already included** in Table 1 (see lines 350–352):
>
> - **COFormer-direct:** Stage 2 only (policy imitation).
> - **COFormer-IL:** Stages 1 + 2 (dynamics pretraining → policy imitation).
> - **COFormer-RL:** Stages 1 + 2 + 3 (dynamics → policy → RL).
>
> Across all 8 COPs in Table 1, *COFormer-IL* consistently outperforms *COFormer-direct*, indicating that the dynamics stage provides a stronger initialization for policy learning and leads to better final solution quality. *COFormer-RL* further improves over *COFormer-IL*, showing that RL fine-tuning adds an additional performance gain on top of imitation learning. Moreover, **Figure 13** reports the training curves of Stage 2 with and without Stage-1 pretraining: starting from a dynamics-pretrained checkpoint consistently accelerates convergence and achieves a lower final imitation loss compared to training Stage 2 from scratch. These results support the usefulness of Stage 1 for both optimization efficiency and final performance.
>
> Regarding the reviewer’s question about “skipping” the dynamics stage, Stage 1 is **not strictly required** for RL to be effective. In the large-scale experiments in **Table 5** (TSP1000 and CVRP1000), to reduce computational cost we omit the dynamics stage and train *COFormer-direct* with Stage 2 only, then apply RL fine-tuning directly on top of this policy (Appendix D.1, lines 1075–1077). Even in this reduced 2-stage setting (Stages 2 + 3), RL still yields consistent improvements over *COFormer-direct* on both TSP1000 and CVRP1000. This suggests that while Stage 1 is beneficial for efficiency and provides modest gains in the small- and medium-scale regime, the combination of **policy imitation + RL fine-tuning (Stages 2 + 3)** is already sufficient to obtain strong performance when computational budget is limited.

---

> ### Author Response · Authors · 2025-12-03
>
> **[Q4]**
>
> We thank the reviewer for raising this question. In COFormer, all continuous features are embedded via a unified **μ-law + binning** scheme: raw values (with different scales across COPs) are first normalized into a fixed range by a μ-law transformation, and then discretized into a fixed number of bins which are mapped to integer token IDs. This allows us to represent heterogeneous continuous fields in a **single numeric token space**.
>
> There is indeed a trade-off in the bin count: fewer bins increase quantization noise and may blur fine-grained numeric distinctions, while more bins increase the size and sparsity of the numeric vocabulary, which can hurt training efficiency. Our default choice of 1,800 bins was selected as a practical compromise between these two effects based on preliminary experiments.
>
> In the revised version, we add **Appendix F** to explicitly study the trade-off between discretization resolution and stability. On 6 routing problems with $N=20$ (TSP, CVRP, OP, PCTSP, SPCTSP, ATSP), using the same architecture and training recipe as *COFormer-direct*, we conducted ablation experiments on μ-law variant and the bin count (from 1,800 up to 6,800 bins). Across all settings, training remains stable and performance differences are modest (typically within **1–2%** on each task), with no consistent monotone gain beyond the default setting. These results indicate that COFormer is **not overly sensitive** to the specific choice of μ-law parameters or bin count, and that our main conclusions do not rely on a finely tuned trade-off between discretization resolution and quantization noise.

---

### Official Review · Reviewer_LeQb · 2025-10-31

**Soundness:** 2
**Presentation:** 2
**Contribution:** 1
**Rating:** 2
**Confidence:** 4

**Summary:**

This paper proposes COFormer, a unified framework based on the Transformer architecture for solving various combinatorial optimization problems. It introduces two core techniques:
CO-prefix, which compactly encodes static problem information through prefix token blocks;
and a three-stage training strategy, which first performs dynamic and policy pretraining via imitation learning, followed by fine-tuning through reinforcement learning.

**Strengths:**

1.	The authors introduce the concept of next-token prediction to handle different combinatorial optimization problems and propose two methods to improve training efficiency. The effectiveness of these methods is validated across 8 combinatorial optimization problems.

**Weaknesses:**

1.	Although the paper claims that its idea is inspired by next-token prediction, it remains unclear how the proposed model fundamentally differs from existing Neural Combinatorial Optimization (NCO) approaches. NCO models also treat nodes as tokens and perform autoregressive next-token prediction using Transformer-like architectures. From the perspective of policy network and training algorithm, the method appears to simply adopt a different Transformer-like model and RL-like algorithm.
2.	Based on the above point, the main difference between COFormer and prior NCO methods appears to lie primarily in how problem inputs are processed and tokenized rather than in the core modeling framework. The paper proposes a unified tokenization strategy that converts heterogeneous problem inputs into sequences of tokens, claiming this enables a single model to handle various CO problems. However, this approach seems to be more of a technical workaround than a conceptual breakthrough. In essence, the heterogeneity issue in existing NCO methods arises because different problems have node features with varying dimensions when each node is treated as a token. A straightforward alternative is to flatten all node features into a one-dimensional sequence and regard each scalar feature as a token, which would already standardize the input dimensionality across different problems. From this perspective, the proposed “tokenization of each scalar” resembles such a flattening operation, and the paper does not clearly articulate what essential advantages COFormer introduces beyond NCO methods with this simple re-encoding trick.
3.	Although COFormer claims to handle diverse combinatorial optimization problems, it remains unclear whether the approach can effectively process problems with edge-level features, such as asymmetric TSP, where the token sequence could become extremely long and computationally expensive.
4.	In the experimental results, COFormer does not show a clear advantage over GOAL [1] in performance. Moreover, several problem settings evaluated in GOAL, such as asymmetric TSP, are not considered in the experiments of this paper, making the claimed generalization ability less convincing.
5.	Existing encoder-decoder NCO models have already achieved the effect of encoding static information only once [2], so the core CO-prefix method is a relatively weak innovation.
6.	There is a lack of ablation experiments separating the CO-prefix and each learning stage.
7.	In imitation learning, the input content for the two stages differs for the same network, but the method explanation, including the definition of the state and action, is unclear.
8.	The reinforcement learning training approach has already been extensively validated in the NCO field. Therefore, the conclusion in Section 4.5 — “The ability to improve performance without external supervision highlights the potential of COFormer as a general-purpose solver for a wide range of COPs.” — is not sufficiently supported.
9.	The experimental results for the COFormer RL sampling method are missing.
10.	The few-shot learning capability is only demonstrated on a few routing problems of the same type and does not include the 8 CO problems in the main results.

[1] GOAL: A generalist combinatorial optimization agent learning. ICLR 2025.

[2] MVMoE: Multi-task vehicle routing solver with mixture-of-experts. ICML 2024.

**Questions:**

Please refer to the weaknesses.

---

> ### Author Response · Authors · 2025-12-03
>
> **[W1/W2]**
>
> We thank the reviewer for the careful comparison with prior NCO work. We agree that, at a high level, COFormer is inspired by the same *next-token prediction* paradigm where an autoregressive policy is learned with Transformer-like architectures. However, we respectfully disagree that COFormer is “merely” a different Transformer with a simple input re-encoding. When the reviewer writes that “NCO models also treat nodes as tokens and perform autoregressive next-token prediction,” this mostly refers to pointer-network–style and AM-style methods [1,2], where *each node corresponds to an embedding* and the Transformer acts as a feature encoder over a graph, so the MDP trajectory is only **implicit** in the sequence of selected node indices. In COFormer, we instead **flatten the entire MDP trajectory** (static problem description, time-varying states, and actions) into a 1D token sequence and treat this explicit trajectory as the primary modeling object, with a **single shared vocabulary** across all COPs. Heterogeneous COPs are thus cast into the *same* representational form, and **differences between COPs are compressed into differences in sequence length and token patterns**, rather than into different node feature spaces. Under this representation, a single Transformer backbone can be trained **without any problem-specific graph encoders or adapters**, and learning shared cross-problem representations.
>
> Wildly  experiments indicate that COFormer indeed learns **shared latent structure** rather than benefiting only from a re-encoding trick. Appendix E.1 visualizes hidden trajectories from six COPs using PCA+UMAP and shows that states from different problems lie on overlapping manifolds, suggesting that the model organizes them by high-level semantics instead of per-problem field order (also see reply to reviewer wAjh [W1/Q1]). Section 4.4 and Appendix E.2/E.3 further show strong few-/zero-shot transfer ability of COFormer. These behaviors, to our knowledge, have not been demonstrated by prior NCO models with simple per-problem flattening, and they provide concrete advantages beyond standardizing input dimensionality.
>
> Finally, regarding the concern that the main difference “lies primarily in how problem inputs are processed and tokenized,” we note that **representation design is itself a core locus of novelty** in many influential sequence-decision works: DT[3] and GATO[4] both use standard Transformer blocks and standard objectives, yet are considered conceptually new because they reframe RL and robotics as sequence modeling via specific trajectory encodings. COFormer follows a similar spirit for combinatorial optimization: rather than significantly modifying the model structure, we change the *representational assumptions* of NCO—from per-problem node feature spaces to a **shared trajectory token space with a global numeric vocabulary**. This representational shift is also recognized by other reviewers: PG5P highlights the idea of building a foundation model for combinatorial optimization via tokenization as novel, while BUAW and wAjh emphasize the CO-prefix and unified cross-problem architecture as key steps toward a general-purpose COP solver.
>
> > [1] Vinyals, Oriol, Meire Fortunato, and Navdeep Jaitly. "Pointer networks." *Advances in neural information processing systems* 28 (2015).
> >
> > [2] Kool, Wouter, Herke Van Hoof, and Max Welling. "Attention, learn to solve routing problems!." *arXiv preprint arXiv:1803.08475* (2018).
> >
> > [3] Chen, Lili, et al. "Decision transformer: Reinforcement learning via sequence modeling." *Advances in neural information processing systems* 34 (2021): 15084-15097.
> >
> > [4] Reed, Scott, et al. "A generalist agent." *arXiv preprint arXiv:2205.06175* (2022).

---

> ### Author Response · Authors · 2025-12-03
>
> **[W5]**
>
> We agree that classical encoder–decoder NCO models already encode static information only once. Our intention is therefore not to claim novelty for the idea of “encoding static data a single time,” but to propose **CO-prefix as a representation mechanism tailored for a unified, multi-COP foundation-style model**.
>
> In encoder–decoder NCO, including recent multi-task VRP solvers **MVMoE**, the encoder operates on **problem-specific graph or sequence features**. To be specific, MVMoE defines the static node features for VRP variants as $S_{i}=(y_{i}, \delta_{i}, e_{i}, l_{i})$ (coordinates, demands, time windows) and the dynamic features as $D_{t}=(c_{t}, t_{t}, l_{t}, o_{t})$ (remaining capacity, current time, route length, open-route flag), and then takes the **union of these VRP attributes** across variants as its shared feature space. MVMoE This yields a powerful *multi-task VRP* solver, but the representation is still tailored to routing-style problems and is not directly designed to cover non-VRP COPs such as knapsack, flowshop scheduling or 3D bin packing.
>
> In contrast, **CO-prefix imposes a stronger, problem-agnostic constraint**: *all* static information—whether it comes from a Euclidean routing graph, a job–machine processing-time matrix, or a 3D bin packing item list—**can be serialized into a linear prefix of tokens drawn from the same global vocabulary**, with no separate graph encoder and no problem-specific adapters. In practice, the static part of each COP is first organized as an arbitrary collection of human-readable key–value tensors (see reply to reviewer wAjh [W1/Q1]), and CO-prefix simply flattens these into the prefix sequence. This key–value–based interface is highly flexible: adding a new COP only requires defining its static fields as key–value pairs and specifying how to serialize them, without changing the backbone architecture.
>
> This design has two practical consequences that go beyond simply “encoding once”: (i) it lets us plug heterogeneous COPs (routing, knapsack, scheduling, 3D bin packing) into **one shared Transformer backbone** by only changing how we serialize the prefix, rather than designing different encoders or feature sets for each COP family; and (ii) it controls sequence length by moving large static structures into a compact, reusable prefix, which is crucial for training the model on extremely large-scale instances such as TSP1000 and CVRP1000.
>
> This problem-agnostic, token-based CO-prefix abstraction is also recognized by other reviewers as a **strength** of the paper: for example, reviewer BUAW describes it as “*elegantly decoupling static problem representations from dynamic sequences*” and “*a step toward truly general-purpose neural optimizers*”, and reviewer wAjh similarly highlights the CO-prefix as an original and practically motivated way to reduce sequence length and enable a unified cross-problem model.

---

> ### Author Response · Authors · 2025-12-03
>
> **[W3/W4]**
>
> We thank the reviewer for raising the issue of edge-level features and the comparison with GOAL. COFormer is not restricted to node-feature problems: in the revised version we explicitly add experiments on **asymmetric TSP (ATSP)** to address this concern, and the detailed CO-prefix / tokenization scheme is provided in **Appendix A.10**. We conduct joint policy learning (stage 2) on ATSP together with five other routing COPs. As shown in Table 9 in Appendix F, COFormer achieves good solution quality **without changing the model context length**, which demonstrates that COFormer can handle dense edge-level features like ATSP in practice and that the token sequence does not become “extremely long and computationally expensive” at the scales considered in this work.
>
> For [W4], we agree that GOAL is a strong and highly relevant baseline. Although the two works cover different COP sets — our benchmark does not fully include all of GOAL’s problem settings, and GOAL likewise does not consider some of the COPs we focus on (such as FFSP and 3DBP) — what matters here is that the **generalization claim of COFormer is architectural**. By design, COFormer introduces much less problem-specific inductive bias, and thus has the potential for **stronger cross-problem generalization ability** than GOAL.
>
> - GOAL is designed as a *graph generalist*, where each problem is represented as a graph with carefully chosen node/edge features and new tasks require dedicated input/output heads and problem-specific inductive biases.
> - COFormer is a *problem-agnostic sequence model*: all COPs are cast into a **shared token space** via CO-prefix and trajectory tokens, and adding a new problem only requires specifying the key–value composition of its CO-prefix and MDP states, without modifying the backbone architecture.
>
> Because COFormer deliberately uses fewer problem-specific inductive biases than GOAL, some loss in raw performance on individual tasks is expected. Nevertheless, on the overlapping routing and knapsack problems COFormer and GOAL are **competitive and trade wins**, which indicates that this more uniform, low-bias design remains effective and reliable in practice. We therefore view the two lines of work as complementary: GOAL pushes architectural sophistication for graph COPs, while COFormer explores how far one can go with a *uniform* next-token modeling interface for heterogeneous COPs, a broader, adapter-free generalization perspective.
>
> In addition, we have demonstrated the generalization ability of COFormer through extensive experiments (Figures 4, 9, and 11), and this has been recognized by other reviewers; for example, reviewer PG5P highlights the extensive cross-problem evaluation and notes that the model shows promising few-shot and even zero-shot adaptation.

---

> ### Author Response · Authors · 2025-12-03
>
> **[W6]**
>
> We appreciate the request for more explicit ablations.  We would like to clarify that CO-prefix is not an independent “plug-in” module, but a change in how the MDP state is represented.  If we “remove” CO-prefix while keeping all information intact, the static data (distance matrices, processing-time matrices, item attributes, etc.) must be moved back into the MDP state at every time step, so that each state again contains the full static problem description.  This representation exactly degenerates to the GATO / DB1-style setting, where the entire state is flattened into a single sequence without a separate static prefix.  Such “no-prefix” baselines are already included in our experiments and consistently underperform COFormer, especially on long-horizon / large-static problems like FFSP and 3DBP, due to much longer sequences and less stable training, and they are also difficult to scale to large instances such as TSP1000 and CVRP1000.
>
> Regarding the learning stages, **Table 1 already provides a stage-wise ablation**:
>
> - **COFormer-direct:** trained with **Stage 2 only** (policy generation via imitation learning);
> - **COFormer-IL:** trained with **Stages 1 + 2** (Dynamics Forward pretraining followed by policy generation);
> - **COFormer-RL:** trained with **Stages 1 + 2 + 3** (Dynamics Forward → imitation learning → RL finetuning).
>
> Across all 8 COPs reported in Table 1, **COFormer-IL generally achieves better solution quality than COFormer-direct**, indicating that pretraining the dynamics model provides a stronger initialization for the subsequent imitation-learning stage and leads to improved optimization performance. Moreover, **COFormer-RL achieves the best greedy decoding performance for all 8 COPs**, confirming the effectiveness of the RL stage.

---

> ### Author Response · Authors · 2025-12-03
>
> **[W7]**
>
> We thank the reviewer for pointing out this potential confusion. Our intention in Section 3.3 is to define a single token-based MDP interface and then apply two imitation-learning stages on top of it with the **same backbone and the same tokenization scheme**, but with different prediction targets. Given an expert tokenized trajectory $\left(\overline{P^{b}},<\mathrm{X}>, \overline{s_0},<\mid>,\overline{a_0},\overline{s_1},<\mid>,\overline{a_1},\dots\right)$ in this token space, the two IL stages differ only in **which tokens are supervised**:
>
> - **Dynamics-pretraining stage (stage 1)**, the input to the network is the CO-prefix and partial trajectory $\left(\overline{P^{b}},<\mathrm{X}>, \overline{s_0},<\mid>,\overline{a_0},\dots,\overline{s_t},<\mid>,\overline{a_t}\right)$, and the model is trained to predict the next **state tokens** $\overline{s_{t+1}}$, loss is applied and optimized only on the state part (i.e. the blue part in Figure 3).
> - **Policy-imitation stage (stage 2)**, the input is the same CO-prefix and partial trajectory $\left(\overline{P^{b}},<\mathrm{X}>, \overline{s_0},<\mid>,\overline{a_0},\dots,\overline{s_t},<\mid>\right)$, but the model is trained to predict the next **action token** $\overline{a_t}$, loss is applied and optimized only on the action part (i.e. the green part in Figure 3).
>
> The details of the tokenization are illustrated in Figure 6. For each COP, both the CO-prefix and the MDP state are represented as key–value pairs, which we serialize into a single flattened token sequence following a fixed schema: within both the CO-prefix and the MDP state we adopt a GATO-style flattening, sorting human-readable keys lexicographically and then concatenating/flattening all value tensors, which yields deterministic key orders for the CO-prefix and state for every problem. The same key order and token schema are used in *both* IL stages.
>
> The per-problem key orders are summarized below (rows are ordered lexicographically by key name):
>
> - CO-prefix key order
>
>   | Knapshot     | FFSP      | 3DBP | TSP      | CVRP      | OP        | PCTSP     | SPCTSP    |
>   | ------------ | --------- | ---- | -------- | --------- | --------- | --------- | --------- |
>   | item_values  | durations | -    | position | demand    | pos_depot | penalty   | det_prize |
>   | item_volumes |           |      |          | pos_depot | pos_node  | pos_depot | penalty   |
>   |              |           |      |          | pos_node  | prize     | pos_node  | pos_depot |
>   |              |           |      |          |           |           | prize     | pos_node  |
>
> - State key order
>
>   | Knapshot      | FFSP          | 3DBP          | TSP              | CVRP             | OP               | PCTSP            | SPCTSP           |
>   | ------------- | ------------- | ------------- | ---------------- | ---------------- | ---------------- | ---------------- | ---------------- |
>   | capacity_left | machine_query | current_state | current_position | capacity         | current_position | current_position | current_position |
>   |               |               |               |                  | current_position | length           | prize2go         | stoc_prize2go    |
>   |               |               |               |                  |                  | prize            |                  |                  |
>
> Within each problem, the ordering of values is induced purely by the **lexicographic sorting of human-readable key names**. Take the CO-Prefix of FFSP and CVRP as an example:
>
> - FFSP has $N\times M$ `durations` tokens in the CO-Prefix.
> - CVRP has $N$ `demand` tokens followed by 2 `pos_depot` tokens and then $2N$ ` pos_node` tokens in the CO-Prefix.
>
> It is important to note that we do **not** attempt to manually align value orders across different COPs beyond the lexicographic key sorting; all remaining structure is learned by the model in the shared token space. Please also see our response to reviewer wAjh [W1/Q1] for additional discussion of the tokenization design and its implications.

---

> ### Author Response · Authors · 2025-12-03
>
> **[W8]**
>
> We agree that the RL training approach itself is not novel and has been extensively explored in the NCO literature. Our goal in Section 4.5 is therefore **not** to claim novelty for the RL algorithm, but to show that, within the proposed unified token-based COFormer framework, the multi-COP model can still be **systematically improved using only environment rewards**, without any additional expert trajectories. As shown in Table 1 and Figure 5, RL fine-tuning leads to **consistent improvements across COPs**, indicating that standard RL remains effective on top of our IL-pretrained, multi-problem backbone. To avoid overclaiming, we have softened the wording at the end of Section 4.5 to: *“The ability to improve performance without external supervision highlights the potential of COFormer.”* This revised sentence emphasizes the practicality and compatibility of COFormer with RL, rather than attributing the broader “general-purpose” claim solely to the RL stage.

---

> ### Author Response · Authors · 2025-12-03
>
> **[W9]**
>
> We thank the reviewer for pointing this out. In the revised version, we have added the results for COFormer-RL with sampling to the experimental section and corresponding tables.
>
>
>
> **[W10]**
>
> We appreciate the reviewer’s concern that our initial few-shot experiments might appear to focus only on “a few routing problems of the same type.” To provide a more systematic picture, we add new analyses and experiments in **Appendix E** that examine whether COFormer learns **shared latent structure** and **transferable representations** beyond a small subset of routing COPs:
>
> - **Visualization of learned representations (Appendix E.1, Figure 10).** UMAP embeddings of latent states for five routing problems and Knapsack show that their trajectories occupy a largely overlapping manifold in the learned representation space, providing direct evidence of shared latent structure across different COP families rather than only within a single routing type.
> - **Cross-family few-shot transfer (Figure 11(a)).** Starting from a multi-routing checkpoint, fine-tuning on **Knapsack** leads to faster convergence and better final performance than training a Knapsack-only model from scratch. This demonstrates that representations learned on routing COPs can be effectively reused for a different family (packing), and offers a concrete explanation of **why COFormer can few-shot adapt to new COPs**.
> - **Within-family transfer and new constraints (Figure 4 and Figure 11(b)).** Within routing, we study transfer across different routing COPs and across **new constraint types**. We observe strong few-shot improvements and non-trivial zero-shot performance on TSP variants whose constraints were not present in the pre-training mix, indicating that COFormer learns a common routing representation that can be adapted to new constraint patterns with limited additional data.
>
> For **FFSP and 3DBP**, under our present setup we do **not** consistently observe clear improvements from initializing with the multi-routing checkpoint compared to training from scratch. Our analysis in Appendix E suggests that these problems induce much longer MDP trajectories; with the current context-length limit, rollouts must be split into shorter subsequences whose distribution differs from full trajectories, which makes it harder for the model to fully exploit cross-task representation sharing. Importantly, FFSP and 3DBP are **still handled by the *same* shared Transformer backbone in the multi-task setting** (Table 1); the limitation lies in the strength of few-shot transfer under constrained context length, not in an inability to incorporate these problems into a unified architecture.
>
> In summary, the extended experiments in Appendix E, together with Figures 4, 9, and 11, show that COFormer **does exhibit meaningful few-shot generalization**: both within the routing family (including new constraint types) and across families (routing → Knapsack).

---

### Official Review · Reviewer_BUAW · 2025-11-01

**Soundness:** 3
**Presentation:** 3
**Contribution:** 2
**Rating:** 4
**Confidence:** 3

**Summary:**

This paper introduces COFormer, a Transformer-based framework designed to tackle diverse combinatorial optimization problems (COPs) within a unified sequence modeling paradigm. The method incorporates three core innovations:

(1) a CO-Prefix representation that encodes static, problem-specific features (e.g., city coordinates or item attributes) into reusable embeddings shared across tasks;

(2) a Hybrid Non-Causal Transformer that applies bidirectional attention to static tokens and causal attention to dynamic state–action sequences, improving inductive bias and efficiency; and

(3) a Three-Stage Training Paradigm comprising a dynamics pretraining stage (supervised learning of environment transitions), a policy generation stage (imitation learning from expert trajectories), and an RL finetuning stage (policy improvement beyond imitation).

Empirical evaluations across multiple benchmark problems demonstrate the potential of COFormer as a general-purpose neural solver capable of handling various COPs within a single architecture, highlighting its promise toward a unified foundation model for combinatorial optimization.

**Strengths:**

1) Unified and Efficient Framework – The proposed CO-Prefix abstraction elegantly decouples static problem representations from dynamic sequences, enabling a single model to generalize across diverse COPs. This is a step toward truly general-purpose neural optimizers.

2) Three-stage training paradigm – The proposed Dynamics Forward → Policy Generation → RL Finetuning pipeline is conceptually well-motivated. It mirrors the classical model-based RL structure — learning environment dynamics first, then imitation, and finally reinforcement fine-tuning.

3) Promising empirical direction – The reported results show that COFormer can outperform prior methods on several small-scale COPs, suggesting potential for broader applicability.

**Weaknesses:**

1) Lack of empirical validation for key components
The Dynamics Forward Stage—one of the paper’s most central contributions—is not ablated or analyzed in isolation.
Without direct comparisons (e.g., COFormer without dynamics vs. full COFormer), it remains unclear whether this stage truly improves optimization performance or merely stabilizes training. This omission significantly weakens the empirical credibility of the claimed contributions.

2) Limited analysis of generalization in the Few-Shot Ability section
It remains unclear how the model’s few-shot capability extends beyond similar routing domains—such as to packing (3DBP) or scheduling (FFSP)—especially given that the proposed Dynamics Forward Stage should, in theory, facilitate more transferable representations across different COP families.

3) Limited experimental coverage and baseline clarity
Table 1 does not include RL fine-tuning results for 3DBP and FFSP, and the reason for this omission is not clearly explained. Additionally, it would be helpful to clarify why MatNet and PCT were not trained with RL to ensure fair and consistent comparison across methods.

**Questions:**

1. Could the authors provide a quantitative ablation isolating the contribution of the Dynamics Forward Stage? What specific improvements (if any) does this stage offer in convergence speed or final solution quality?

2. Has the model demonstrated cross-task transfer, e.g., training on routing and testing on scheduling or packing problems?

3. Why were RL fine-tuning experiments for 3DBP and FFSP omitted, and why are some baselines trained differently? Were there technical limitations or theoretical reasons?

4. From Table 5, COFormer-direct-greedy and COFormer-RL-greedy exhibit only marginal differences, which raises doubts about whether the current training paradigm scales effectively to larger instances. Could the authors clarify whether the proposed framework remains computationally and performance-wise effective as problem size grows, and also provide more evidence on how it performs across different tasks such as CVRP and FFSP?

---

> ### Author Response · Authors · 2025-12-03
>
> **[W1/Q1]**
>
> We thank the reviewer for this suggestion. The ablation over the three training configurations **is already included** in the main experiments (see lines 350–352 and Table 1):
>
> - **COFormer-direct:** trained with **Stage 2 only** (policy generation via imitation learning);
> - **COFormer-IL:** trained with **Stages 1 + 2** (Dynamics Forward pretraining followed by policy generation);
> - **COFormer-RL:** trained with **Stages 1 + 2 + 3** (Dynamics Forward → imitation learning → RL finetuning).
>
> Across all 8 COPs reported in Table 1, **COFormer-IL generally achieves better solution quality than COFormer-direct**, indicating that pretraining the dynamics model provides a stronger initialization for the subsequent imitation-learning stage and leads to improved optimization performance. In addition, **Figure 13** reports the training curves of the policy generation stage with and without Dynamics Forward pretraining: starting from a Stage-1-pretrained checkpoint **consistently accelerates convergence** and results in a **lower final training loss** compared to training Stage 2 from scratch.
>
> Taken together, the results in **Table 1** and **Figure 13** provide quantitative evidence that the Dynamics Forward Stage not only stabilizes training but also improves both convergence speed and final optimization performance. We will clarify this ablation more explicitly in the revised manuscript

---

> ### Author Response · Authors · 2025-12-03
>
> **[W2/Q2]**
>
> We appreciate the reviewer’s question on how far COFormer’s few-shot ability and representation sharing extend beyond closely related routing problems.
>
> As the reviewer noted, achieving **semantic unification across heterogeneous COPs** is highly challenging, since different problems can vary significantly in both formal structure and optimization objectives. Many existing works therefore relax this goal in one of two ways:
>
> 1. **Single-domain unification.** Some methods focus on a single problem family (typically routing) and design a unified encoder within that family [1,2], without addressing fundamentally different COP families such as packing or scheduling.
> 2. **Problem-specific adapters.** Some methods introduce problem-specific encoders or adapters for each COP, such as GOAL [3], where each problem type is handled by a dedicated feature extractor instead of a fully shared representation space.
>
> In contrast, **COFormer deliberately minimizes problem-specific inductive bias**: the CO-Prefix representation and shared Transformer backbone operate on a unified token space across all COPs. Our goals are twofold:
>
> 1. Through joint multi-task training, a single model can **solve all COPs covered in the training set** with one architecture and parameter set (as shown in Table 1).
> 2. A checkpoint trained on a subset of COPs can be **quickly adapted via fine-tuning** to new COPs that are at least in the same problem family as one of the training tasks (as illustrated in Figure 4).
>
> Beyond this within-family setting, we also observe **effective cross-family transfer**. In Appendix E, we add new analyses that visualize **semantic representations across COPs** and evaluate few-shot transfer:
>
> - **Representation visualization (Figure 10).** UMAP plots of latent states for 5 routing problems and Knapsack show that their trajectories occupy a largely overlapping manifold in the learned representation space, suggesting that COFormer captures shared structural regularities rather than merely memorizing problem-specific patterns.
> - **Cross-family transfer (Figure 11(a)).** Starting from a **multi-routing checkpoint**, we fine-tune on **Knapsack**. The routing-pretrained model converges faster and achieves better final performance than a Knapsack model trained from scratch, indicating that representations learned on routing tasks transfer effectively to a different COP family (packing).
> - **Within-family transfer (Figures 4 and 11(b)).** Within the routing family, we study transfer across different routing COPs and across constraints. These results show strong few-shot improvements and non-trivial zero-shot performance, demonstrating that shared representations learned on some routing problems can be effectively reused to solve other routing COPs.
>
> For **FFSP and 3DBP**, initializing from the multi-routing checkpoint **does not yet yield a clear advantage** over training from scratch. Our preliminary analysis suggests that these problems induce **much longer MDP trajectories**; under the current context-length limit, rollouts must be split into short subsequences whose distribution differs from full trajectories, which likely encourages more problem-specific representations. We explicitly acknowledge this limitation in Appendix E and plan to explore longer-context architectures and trajectory-aware segmentation as future work.
>
> In summary, our current experiments show that COFormer **does achieve meaningful cross-task transfer**: it supports both within-family transfer (across routing COPs and constraints) and cross-family transfer (routing → Knapsack).
>
> > [1] Federical Berto et al., RouteFinder: Towards Foundation Models for Vehicle Routing Problems, ICML Workshop, 2024.
> >
> > [2] Zhou, Jianan, et al. "Mvmoe: Multi-task vehicle routing solver with mixture-of-experts." arXiv preprint arXiv:2405.01029 (2024).
> >
> > [3] Drakulic et al., Goal: A Generalist Combinatorial Optimization Agent Learning, 2025.

---

> ### Author Response · Authors · 2025-12-03
>
> **[W3/Q3]**
>
> We thank the reviewer for pointing out the ambiguity around RL training in Table 1.
>
> **(a) MatNet and PCT.**
> We apologize for the lack of clarity in the original text. Both **MatNet** and **PCT** are *RL-based* solvers in their original papers, and in our experiments we follow their official implementations and training protocols, i.e., they **are trained with RL** rather than purely supervised learning. We will explicitly state this in Section 4.2 to avoid confusion and to make the comparison protocol across methods clearer.
>
> **(b) RL fine-tuning for 3DBP and FFSP.**
> We did not include RL fine-tuning results for **3DBP** and **FFSP** mainly due to technical limitations of our current batched environment implementation. Unlike the other COPs, these two problems have more complex environment transition logic (e.g., machine scheduling and 3D packing constraints), and we were not able to build a reliable *batched* simulator; as a result, rollouts for 3DBP and FFSP can only be generated **serially**. This leads to extremely low sample throughput and makes RL training on these tasks prohibitively inefficient and unstable under our computational budget. Rather than reporting very noisy or under-trained RL results, we chose to present only the imitation-learning performance for these two COPs. We will clearly acknowledge this as a limitation in the revised manuscript.
>
> We emphasize that RL fine-tuning on the remaining COPs (TSP, CVRP, OP, PCTSP, SPCTSP, Knapsack, etc.) consistently improves solution quality, as shown in **Figure 5** and the large-scale experiments in **Table 5**, which we believe provides strong evidence for the effectiveness of the proposed framework.

---

> ### Author Response · Authors · 2025-12-03
>
> **[Q4]**
>
> We thank the reviewer for raising the concern about scalability. In the revised manuscript, **Table 5 now reports large-scale results on both TSP1000 and CVRP1000**, and we clarify that the apparently “marginal” difference in the *Score* metric actually hides non-trivial improvements in solution quality.
>
> - On **TSP1000**, *COFormer-direct-greedy* and *COFormer-RL-greedy* achieve Scores of 99.54% and 99.64%, respectively, both very close to the expert baseline (100%). However, in terms of optimality gap, RL still brings a clear benefit: the gap is reduced from **9.63% → 7.62%** for greedy decoding (≈20% relative reduction), and from **4.86% → 3.97%** for sampling.
> - On **CVRP1000**, the effect is even more pronounced. *COFormer-direct-greedy* has a gap of **5.88%**, whereas *COFormer-RL-greedy* reduces it to **3.48%** (≈41% relative reduction). With sampling, the gap further decreases from **2.50% → 1.41%** (≈44% relative reduction), yielding the best overall performance (Score 99.89%).
>
> Thus, although the *Score* metric saturates near 100% on these large instances, the RL stage still consistently improves the optimization objective, especially for the more challenging CVRP1000 case. From a computational perspective, the CO-Prefix mechanism significantly **reduces** sequence length: the token sequences of COFormer are orders of magnitude shorter than those required by GATO. These results indicate that the proposed training paradigm **remains both computationally feasible and performance-effective as the problem size grows to N = 1000** on two different tasks.

---

### Meta-Review · Area_Chair_Q22i · 2026-01-06

**Summary:**

This paper proposes a Transformer-based framework that aims to address a wide range of combinatorial optimization problems under a unified sequence modeling paradigm.

The reviewers generally agreed on the following strengths:
- The idea of a unified sequence modeling paradigm for combinatorial optimization is interesting.
- The training pipeline is well structured and well-motivated.
- The empirical results are promising, at least on small-scale benchmarks.

However, the reviewers raised several concerns, including:
- W1. Conceptual similarity with existing approaches (partially addressed).
- W2. Lack of clarity and technical detail, which hinders a deeper understanding of the method (partially addressed).
- W3. Missing recent baselines (unaddressed).
- W4. Lack of ablation studies and evaluations on larger-scale datasets (addressed during the rebuttal).

The paper received one positive and three negative reviews. Importantly, the positive reviewer explicitly stated he/she is not willing to champion the paper and also stated that the raised concerns are fair points.

While the authors made efforts during the rebuttal to address W1 and W2, as far as the AC can see, the core issues are not fully addressed. Moreover, given the limited overall support for the paper, the work would benefit from another round of reviews.

**Reviewer Concerns:**

W1 and W2 are only partially addressed.
W3 remains unaddressed.
W4 is addressed.

**Reviewer Scores:**

While reviewers may adjust their scores slightly, it is unlikely that the paper will receive sufficient support for acceptance.

---

### Decision · Program_Chairs · 2026-01-26

Reject